

# Global and regional estimates of warm cloud droplet number concentration based on 13 years of AQUA-MODIS observations

Ralf Bennartz[1,2], John Rausch[2]

[1] Earth and Environmental Science Department, Vanderbilt University, Nashville, TN, 37240, USA
[2] Space Science and Engineering Center, University of Wisconsin – Madison, WI, 53706, USA

*Correspondence to*: Ralf Bennartz (ralf.bennartz@vanderbilt.edu)

**Abstract.** We present and evaluate a climatology of cloud droplet number concentration (CDNC) based on 13 years of Aqua-MODIS observations. The climatology provides monthly mean 1x1 degree CDNC values plus associated uncertainties. All values are in-cloud values, that is, if the grid box is covered to 10% with clouds, then the reported CDNC value will be valid for the cloudy part of the grid-box. Here, we provide an overview on how the climatology was generated and assess and quantify potential systematic error sources including effects of broken clouds, and remaining artefacts caused by the retrieval process or related to observation geometry. Retrievals and evaluations were performed at the scale of initial MODIS observations (in contrast to some earlier climatologies, which were created based on already gridded data). This allowed us to implement additional screening criteria, so that observations inconsistent with key assumptions made in the CDNC retrieval could be rejected. Application of these additional screening criteria led to significant changes in the annual cycle of CDNC both in terms of its phase and magnitude. After an optimal screening was established a final CDNC climatology was generated. Resulting CDNC uncertainties for the climatology are in the order of 30% in the stratocumulus regions and 60% to 80% elsewhere. The climatology is available in Network Common Data Format (netCDF) and adheres to the Climate and Forecast (CF) convention. The climatology is available via Digital Object Identifier (Bennartz and Rausch, 2016, 10.15695/vudata.ees.1).

## 1 Introduction

Presently one of the largest sources of uncertainty in predicting future climate is the degree to which clouds will alter the Earth's radiative balance (Stocker et al., 2013). In particular stratiform boundary layer clouds play an important role in modulating earth albedo and are 'at the heart of tropical feedback uncertainties in climate models' (Bony and Dufresne, 2005). Aerosol-cloud interactions contribute to these uncertainties, e.g. through the First Indirect Aerosol Effect (Twomey, 1974), in which an aerosol-perturbed cloud reflects solar radiation more efficiently than an unperturbed cloud under otherwise identical conditions.

Satellite observations play a critical role in understanding current-day variability of clouds, in validating and constraining climate models, and in furthering our understanding of cloud processes. Based on earlier work (Brenguier et al.,





2000;Schuller et al., 2005;Schuller et al., 2003), Bennartz (2007) published an initial version of a Cloud Droplet Number Concentration (CDNC) climatology for liquid boundary layer clouds using cloud parameters derived from NASA's Aqua Moderate Resolution Imaging SpectroRadiometer (MODIS). Revised and extended versions of this climatology have been used for climate model validation and evaluation (Storelvmo et al., 2009;Hoose et al., 2008;Hoose et al., 2009;Makkonen et

al., 2009;Zhang et al., 2012a;He et al., 2015;Wang et al., 2015;Ban-Weiss et al., 2014). A variety of observational case studies and process studies were also published using similar approaches for deriving CDNC (Boers et al., 2006;George and Wood, 2010;Painemal and Zuidema, 2010;Rausch et al., 2010;Bennartz et al., 2011). Various authors have addressed shortcomings and issues related to CDNC climatologies (Merk et al., 2016;Grosvenor and Wood, 2014) as well as issues related to the cloud-retrievals underlying the CDNC climatologies (Zhang and Platnick, 2011;Nakajima et al., 2010;Maddux

et al., 2010;Horvath et al., 2014). Painemal and Zuidema (2011) validate MODIS-derived CDNC against in-situ observations taken in the South Pacific during VOCALS-Rex (Wood et al., 2011). Wood et al. (2012) point out important other factors, namely precipitation generation, that hamper the interpretation of CDNC results in correlative observational studies on the context of the first indirect aerosol effect. Several studies also propose different and potentially more elaborate approaches for deriving CDNC. For example, Rosenfeld et al. (2012) propose a spatially highly-resolving satellite mission dedicated to

the retrieval of cloud condensation nuclei for convective clouds and also point to limitations of the CDNC retrieval approach pursued here when applied to convective clouds. Zeng et al. (2014) propose a combination of Cloud-Aerosol Lidar with Orthogonal Polarization (CALiOP) observations with MODIS observations to retrieve CDNC.

In light of these developments, the objective of the present paper is threefold. Firstly, we wish to update older versions of the CDNC climatology by Bennartz (2007) accounting for the 13 full years Aqua-MODIS observations available by now, taking

advantage of the revised and improved NASA MODIS Collection 6 cloud retrievals as outlined in Platnick et al. (2015), with better traceability via DOI-assignment. Secondly, we wish to provide readers and users of the climatology with a comprehensive overview on issues and limitations of the climatology. Thirdly, wish to quantitatively assess possible systematic error sources, quantify uncertainties, and provide validation of the climatology.

Systematic errors and uncertainties in the CDNC climatology stem largely from assumptions made in the retrieval process

and in the cloud model that underlies the CDNC retrievals. In particular three assumptions are critical throughout the retrieval process:

1. The cloud is assumed to be horizontally homogeneous. This is an assumption frequently made in various cloud retrievals, not only CDNC retrievals. It potentially leads to systematic errors for situations, where the observed scene deviates from horizontal homogeneity.

2. Cloud liquid water content (LWC) is assumed to increase linearly from cloud base to cloud top. The rate of increase of LWC with height (in $kg/m^4$) is often called the 'condensation *rate*' similar to the use of the term 'lapse *rate*' for the change of temperature with height. The assumption of linearity has been a matter of some confusion in the context of CDNC retrievals. We point out that it is not necessary to hold the condensation rate constant at its maximum adiabatic value. A strict linear assumption is also not necessary as long as reasonable assumptions about



the vertical LWC profile are made. For example, Boers et al. (2006) relax the strict linear assumption near cloud top to better account for cloud top entrainment.

3. CDNC is typically assumed constant throughout the cloud's vertical extent. This assumption can also be relaxed somewhat as long as a clear dependency of CDNC on height above cloud base exists.

In the past, we have labelled these three assumptions collectively as the 'adiabatic cloud model'. This labelling appeared justified because the above-described assumptions are inspired by an idealized air parcel rising under moist adiabatic conditions. However, here we refrain from using the term 'adiabatic cloud model' as it has led to misunderstandings mostly revolving around the usage of the word 'adiabatic'. Indeed, the cloud model described above is not strictly adiabatic as it allows for modification and deviations from strict adiabaticity in terms of the profile of both, LWC and CDNC. A term that

probably better, albeit less attractively, describes the intent of the above assumptions would be the 'Idealized Stratiform Boundary Layer Cloud' (ISBLC) model, which we will use throughout this paper. In fact, most authors use sub-adiabatic profiles and condensation rates around 80% of their maximum value are often found in experimental studies (Wood, 2012). A recent study by Merk et al. (2016) finds LWC at about 75 % of its adiabatic value in updrafts and about 60 % of its maximum adiabatic value in downdrafts.

While the ISBLC captures important aspects of actual stratiform boundary layer clouds, none of its assumptions will ever be fully valid for any observed cloud. The true three-dimensional variability of clouds poses significant challenges to any remote sensing algorithm and a growing body work has been devoted toward understand the impact of this variability on remote sensing estimates of CDNC. Furthermore, the cloud microphysical interpretation of retrieved CDNC is also not straight forward, as ultimately one would be interested in the number of cloud droplets activated at cloud base and not the

number of cloud droplets observed. Entrainment mixing processes, precipitation formation, and additional activation above cloud base can lead to differences between these two properties. We will elaborate more on issues related to three-dimensional cloud structure as well as cloud microphysical assumptions in Section 3.

The remainder of this paper is structured as follows. In Section 2 we briefly describe the datasets as well as the principal methods used here to derive CDNC from satellite observations. Section 3 summarizes issues related to the use of the ISBLC

as well as other assumptions made in the retrieval process, thereby providing interpretational context for the use of satellite-derived CDNC retrievals as well as guidance on the expected magnitude and relative importance of uncertainties introduced by the various assumptions. In Section 4 we address actual uncertainties and possible biases in CDNC retrievals. Potential biases are for example caused by remaining artefacts in the underlying MODIS retrievals caused by unresolved dependencies on observation geometry or underlying assumptions on the width of the droplet spectrum. In addition, in Section 4 we also

validate our CDNC retrievals against in-situ observations of CDNC taken during the VOCALS-Rex campaign and summarized by Painemal and Zuidema (2011). In Section 5 we evaluate the 13-year climatology of MODIS observations. In our analysis we put particular emphasis on the phase and amplitude of the observed annual cycle of CDNC over various





regions of the globe. We further identify areas where trends in CDNC are observed. In Section 6 we provide concluding remarks and discussion of remaining issues that could help improve future satellite-based CDNC estimates.

## 2. Datasets and Methods

### 2.1. MODIS Collection 6 cloud parameters

Observational data are from NASA's MODIS Collection 6 (C6) Level-2 Cloud Product (Platnick et al., 2015). The term 'Level-2' refers to individual MODIS cloud retrievals at a resolution of 1x1 km at nadir. The Level-2 Cloud Product provides retrievals of cloud optical thickness, cloud top temperature, and three droplet effective radii retrievals using observations at 1.6, 2.1 and 3.7 μm. These cloud parameter retrievals form the basis of our CDNC retrievals. Relative to the earlier MODIS Collection 5, there are several improvements in retrievals of parameters necessary to determine CDNC.

These improvements include a better co-registration of the visible and near-infrared focal planes of the Aqua-MODIS instrument as well as significant improvements in the forward radiative transfer models used in the retrieval framework (Platnick et al., 2015). The impact of these changes on CDNC retrievals and gridded climatologies is assessed in detail in Rausch et al. (2016). For the current study, Level-2 cloud retrievals are from Aqua MODIS spanning the years 2003 through 2015. Additionally, for some comparisons with in-situ observations, selected Terra-MODIS granules were used also.

### 2.2. Derivation of CDNC

Under the ISBLC assumption, closed formulas can be derived that relate between two different pairs of cloud physical variables. The first pair of variables are cloud optical depth and effective radius at cloud top, and the second pair of variables are CDNC and cloud geometrical thickness (e.g. Brenguier et al. (2000)). Following the notation of (Bennartz, 2007), these relations are:

$$W = \frac{5}{9}\rho_l \tau r_{e,top} = \frac{1}{2}c_w H^2 \qquad (1)$$

where $W$ is the liquid water path, $\rho_l$ the density of liquid water, $\tau$ the optical depth, $r_{e,top}$ the effective radius at the top of the idealized ISBLC, $c_w$ the condensation rate, and $H$ the ISBLC's geometrical thickness. The second relation provides an

estimate for CDNC:

$$N = \frac{\tau^3}{k}\left[2W\right]^{-5/2}\left[\frac{3}{5}\pi Q\right]^{-3}\left[\frac{3}{4\pi\rho_l}\right]^{-2}c_w^{1/2} \qquad (2)$$





where N represents CDNC, $k$ is a factor related to the dispersion of the assumed cloud droplet size distribution, and $Q$ is the scattering efficiency of the cloud droplets. The variable k exhibits some variability (Brenguier et al., 2011; Martin et al., 1994) but is set constant at a value of $k=0.8$ here. Similarly, $Q$ is set to its geometric optics limit value of $Q=2$. Bennartz (2007) shows that uncertainties in the representation of $Q$ and $k$ are only a minor contributor to the total uncertainty in $N$ and

$H$.

The above equations allow for a conversion between ($\tau, r_{e,top}$) and (N, H) thereby enabling the use of NASA's retrieved effective radius and optical depth for calculating CDNC without the need for dedicated retrieval algorithms that would have to be build on Level-1 reflectances. An important assumption made in this conversion between ($\tau, r_{e,top}$) and ($N, H$) is that the retrieved effective radius is valid at the top of the ISBLC, as indicated by the subscript 'top' in the above equations.

However, satellite-derived effective radii are typically valid at some penetration below cloud top that depends on observation wavelength and geometry. Thus, if the satellite-derived effective radius is used directly in the above conversion, one assumes that $r_{e,top} = r_{e,retrieved}$. Throughout this paper we employ this assumption and use the effective radius retrieved at 3.7 μm to calculate CDNC. The implications and limitations of this assumption are discussed further in in Section 3.

### 2.3 Gridded Climatology

The individual CDNC retrievals discussed above were used to calculate global fields of monthly averages of CDNC on a regular 1°x1° latitude-longitude grid for the years 13 years 2003 - 2015. First, daily averages were calculated as the arithmetic mean of all Level-2 MODIS-retrieved CDNC values within each 1°x1° grid box within one day. In order to provide a valid daily 1x1 degree average at least ten valid Level-2 retrievals had to be within that grid-box. Screening criteria performed on individual pixels are outlined further below. If, for a given month and grid-box, more than ten days had valid

daily values, the arithmetic mean of those was assigned to be the monthly mean value. The so-derived monthly mean fields, along with uncertainty estimates, are part of the published dataset.

Uncertainty estimates were generated as follows: In addition to the daily mean value of CDNC for each 1°x1° grid box, the variance of Level-2 CDNC observations within this grid-box was also calculated. The monthly uncertainty for this grid-box was then calculated as the square-root of the mean of the daily variances over the course of each month, thereby assuming

uncertainties between days to be uncorrelated. This approach of creating monthly uncertainties is similar to the approach used in NASA's Level-3 gridded MODIS cloud product (Hubanks et al., 2016). The difference between our approach and that of NASA is that we calculate the underlying daily grid-box-level uncertainties from spatial standard deviations of retrieved CDNC whereas NASA uses a-posteriori retrieval errors as basis for their daily grid-box-level uncertainties. We address the difference between these two choices with respect to CDNC in Section 4.

In order to ensure that only valid cloud observations were accumulated, a series of screening criteria were used on each Level-2 data point. The impact of different screening choices is discussed in Section 4. Here we list the final set of screening choices made in the climatology:





1. Both the infrared-derived and the visible/near-infrared-derived cloud phases had to indicate a water cloud.

2. The retrieved cloud top temperature had to be between 268 and 300 K.

3. The cloud mask had to indicate the observation to be cloudy but not over ice or land.

4. Clear-sky-restoral and pixels identified in the MODIS Level-2 cloud product as partly cloudy pixels were not included.

5. All three effective radii retrieved at 1.6, 2.1, and 3.7 μm, respectively, as well as the three corresponding retrieved optical thicknesses had to be valid.

6. Observations were only considered, if the three MODIS-retrieved effective radii stacked up as $r_{e,3.7} > r_{e,2.1} > r_{e,1.6}$, as observations violating this criterion will also violate the key assumption of a vertically increasing LWC in the ISBLC. More details on the motivation of this screening criterion are provided subsequently in Section 3.

## 3. Assumptions and limitations of the ISBLC

NASA's MODIS effective radius and optical thickness retrievals build on the use of 'Nakajima-King Diagrams' (Nakajima and King, 1990) similar to the red mesh shown in Figure 1. The retrieval algorithms effectively map between the observed reflectances, on the x- and y-axis, and the retrieved ($\tau, r_{e,retrieved}$) parameters shown in the red mesh. Details on the actual retrievals used in MODIS C6 can be found in Platnick et al. (2015). Note, that NASA's operational retrievals, as most other retrievals, are performed using a cloud model in which effective radius and liquid water content are constant vertically, thereby replacing assumptions 2 and 3 of the ISBLC while maintaining its assumption 1. This cloud model is sometimes tagged the Vertically Homogeneous Cloud (VHC) model. The ISBLC-based relation between (*N, H*) and the observed reflectances is also depicted in Figure 1 (blue mesh).

Subsequently, we discuss several issues related to the retrieval of cloud optical properties in general and CDNC in particular. All radiative transfer simulations underlying the discussions in this section were performed using the Spherical Harmonics Discrete Ordinate Method (SHDOM) model (Evans, 1998). Spectral optical properties for all aerosols were taken from the Optical Properties of Aerosols and Clouds (OPAC) database (Hess et al., 1998). Simulations were performed for idealized situations over a black surface, nadir observations and a solar zenith angle of 56 degrees. The resulting scattering angle of 124 degrees samples the phase function in a relatively smooth region outside of the backscatter or rainbow peaks. Retrievals on simulated data were performed using Levenberg-Marquardt optimization.

### 3.2. Penetration depth

Figure 2 shows the penetration depth of radiation into ISBLCs as relevant for effective radius retrievals. Firstly, a series of reflectances for ISLBCs with different CDNC and geometrical thickness were calculated. The calculations were used as 'observations'. Secondly, these 'observations' were input to retrieval algorithms using the VHC model (red meshes in Figure





1). We keep the quotation marks around 'observations' here to indicate that this is a simulation experiment. The simulated retrieval process is analogue to the actual MODIS effective radius retrieval where VHC retrievals are applied to stratiform boundary layer clouds that are typically vertically stratified. The so-retrieved effective radius is then compared to the effective radius profile of the ISBLC that underlie the 'observations'. The penetration depth is then defined as the height

below cloud top, where the retrieved effective radius equals the actual effective radius of the underlying ISBLC. The definition of penetration depth used here cannot be considered directly as the radiative penetration depth at a given wavelength because it uses two-channel effective radius retrievals to determine where vertically the retrieved effective radius is representative. For a full discussion of the radiative penetration depth we refer to Platnick (2000), who also discusses in detail the difference between our definition and the radiative penetration depth as well as angular dependencies.

From Figure 2 one can identify that at 1.6 μm the penetration depth is about two to three times higher than that at 3.7 μm. The results are in good general agreement with the results shown in Platnick (2000). See e.g. Table 4 therein. The resulting retrieved effective radii at the three wavelengths are typically within about 1-2 μm of each other with the 3.7 μm effective radius being the largest. Actual clouds of course deviate often substantially from ISBLCs. Some issues related to cloud inhomogeneities are discussed next.

**3.3. Cloud inhomogeneities**

The liquid water content profile does often not increase linearly with height above cloud base and CDNC as well as effective radius might be reduced near cloud top. In particular inhomogeneous mixing processes reduce CDNC and are observed frequently in boundary layer clouds (Burnet and Brenguier, 2007). Furthermore, horizontal inhomogeneities have a critical impact on cloud optical property retrievals. Such artefacts in cloud retrievals associated with inhomogeneous sub-pixel cloud

cover have discussed in detail for example in Zhang and Platnick (2011) and Horvath and Gentemann (2007) among others. Drizzle processes also significantly alter cloud horizontal, vertical structure, and droplet number concentration (Wood, 2012, 2005). Drizzle might also contribute directly to the observed reflectances thereby increasing the retrieved effective radius at 1.6 μm (Suzuki et al., 2011). Typically, most of these issues affect the effective radius at 1.6 μm and 2.1 μm more strongly than the effective radius at 3.7 μm (Zhang et al., 2012b).

The blue line depicted in Figure 1 highlights the issue. Even without accounting for the three-dimensional radiative transfer, partly cloudy scenes will exhibit positively biased effective radii. The blue cross in Figure 1 corresponds to an ISBLC with $N=250$ cm$^{-3}$ and $H=300$ m. The blue line gives the resulting observed reflectance, if this cloud covers between 0 and 100 % of the sensor's field of view. For all three wavelengths the blue line cuts through the retrieval grid toward larger effective radii and smaller CDNC. Figure 3 (right panels) shows the impact on retrieved effective radii as one moves from completely

cloudy to nearly cloud-free. For sub-pixel fractions of open water above 10% the retrieved effective radius at 1.6 μm starts to exceed 2.1 and 3.7 μm but all three effective radii increase significantly as the pixel becomes less cloud-filled. Zhang et al. (2012b) and Hayes et al. (2010) discuss this effect in more detail, which is also present in case of sub-scale inhomogeneities of fully cloud covered observations (i.e. if cloud thickness and liquid water path varies within the field-of-view). Shading



and side-illumination of broken clouds as well as true three-dimensional radiative transfer effects modify this picture somewhat but at the same time the inherent averaging performed over a 1 x 1 km$^2$ MODIS field-of-view averages out some of these higher-order effects (Zhang et al., 2012b). Because of the strong impact of sub-scale inhomogeneity we have limited the climatology to cases where $r_{e,3.7} > r_{e,2.1} > r_{e,1.6}$ , so that in the example shown in Figure 3 (right panels) the largest part

of the retrievals would be (correctly) rejected. We discuss the impact of this criterion on the actual climatology in Section 4.

### 3.4. Aerosol above clouds

Another important topic related to effective radius and CDNC retrievals is the impact of aerosol above clouds. This has received a significant amount of attention in the literature as especially off South Africa biomass burning aerosols, consisting of black carbon (soot), organic compounds, ammonium, nitrate, and sulfate, are often vertically displaced above the

stratocumulus clouds by several kilometres (Wilcox et al., 2009;Painemal et al., 2014;Wilcox, 2012;Haywood et al., 2004). We illustrate the effect of aerosol layers on retrievals in Figure 1 (see golden curve with large dot at the end). We start from the blue cross in Figure 1, which corresponds to an ISBLC with N=250 cm$^{-3}$ and H=300 m. From there we increase aerosol load above the cloud from zero to an optical depths at 550 nm of one toward the golden dot. The aerosol type we choose is 'continental polluted' from the OPAC database (Hess et al., 1998), which has roughly similar optical properties as the aged

biomass burning aerosol reported in Haywood et al. (2003). The corresponding retrievals are shown in Figure 3 (left panels). As aerosol optical depth increases, the retrieved effective radius increases and, not shown, the retrieved optical depth decreases. This results in a decrease in CDNC with increasing aerosol optical depth. The impact of aerosol above clouds depends highly on its single scattering albedo. The more absorbing the aerosol is, the stronger the effect. Note that, unlike for broken clouds, the retrieved effective radius at 3.7 μm remains larger than the 2.1 and 1.6 μm effective radii. Therefore, the

criterion $r_{e,3.7} > r_{e,2.1} > r_{e,1.6}$ does not help reject potentially aerosol-affected observations. Note further, that the effect of aerosol on effective radius can also lead to a decrease in effective radius with increasing aerosol optical depth, depending on observation geometry and where exactly in the Nakajima-King diagrams the observation lies.

### 4. Sensitivity studies

In this section we study the sensitivity of the climatology to choices made in the underlying retrievals as well as data

screening choices, such as the aforementioned criterion $r_{e,3.7} > r_{e,2.1} > r_{e,1.6}$ . We further investigate dependencies of the retrieved CDNC and effective radii on scattering angle and sunglint angle and provide a discussion of uncertainties derived alongside with the CDNC climatology. Lastly, we evaluate the climatology against the limited set of in-situ CDNC observations that were readily available to us. Except for the comparisons to in-situ observations all sensitivity studies were done by re-deriving and stratifying one full year (2008) of MODIS data by different quantities, such as the scan angle or

position within the scan.





### 4.1. Impact of observation geometry

Various studies have already shown that scan dependent biases existed in the older Collection 5 version of MODIS. For example, Maddux et al. (2010) find significantly larger effective radius retrievals along with lower optical depth retrievals near the scan edge than in the centre of the scan. Grosvenor and Wood (2014) address issues the dependency of cloud

microphysical retrievals on solar zenith angle. Issues related to the observation geometry can in principle stem from two different sources. Firstly, as the solar or observation zenith angle increase, the potential effects of side-illumination and shading become more pronounced and, for increasing observation zenith angle, also the field-of-view gets larger. Secondly, assumptions and constraints of the retrieval itself might cause artefacts. For example, the choices made on whether to include sun-glint affected areas might cause differences or the discretization of the retrieval to particular combinations of observation

geometries can cause artefacts in the retrieval. When creating climatologies based on a large number of observations some of these artefacts might average out, whereas others might cause systematic biases also in the climatologies. Subsequently, we first stratify results by various quantities to identify and discuss potential artefacts. Then, we study the impact of these artefacts on the climatology.

Figure 4 shows the dependency of effective radius, optical depth, and CDNC retrievals on both scattering angle and sunglint

angle. Results indicate averages over all valid retrievals for the year 2008 applying the screening criteria corresponding to 'Stratified' and 'Non-Stratified'. One can identify a variety of different artefacts in the dataset. In all cases the 'Stratified' results (blue) show the artefacts more strongly than the 'Non-stratified' cases. We suspect this is caused by the generally larger variability seen in the 'Non-stratified' cases caused by the issues discussed in Section 3 (e.g. horizontal inhomogeneity). Observations affected by e.g. inhomogeneity are at least partly screened out in the 'Stratified' dataset,

which therefore shows additional artefacts more strongly. Fro Figure 4 the following issues can be identified:

- Near the rainbow peak (scattering angle around 145 degrees) and the backscatter peak (Scattering angle 180 degrees) the average retrieved effective radii vary in a manner the roughly resembles the phase function of typical droplet spectra. For these two regions the optical depth is comparably flat with much less variation than the effective radius. We speculate these artefacts could be caused either by the treatment of single scattering in the

MODIS C6 retrievals or by deviations between the spectral widths of the droplet spectra used in the retrievals and the real world droplet spectra. The "hybrid discretization scheme" used in MODIS C6 retrieval is outlined in Platnick et al. (2015), see Table 2.9-2 therein. While the single scattering contribution is calculated exactly at the angles provided in the LUT, larger interpolation errors appear to occur in the rainbow and backscattering region (Platnick et al. (2015), see Figure 2.9-3, lower right panel), potentially consistent with the features found here. The

second possible source of deviations, potential differences between observed and simulated droplet spectral width, could potentially lead to a similar behaviour as it would affect the scattering phase function also most strongly near the rainbow and backscatter peak. We note here that a similar behaviour in retrieved effective radii is also apparent at 1.6 and 2.1 μm (not shown).





- At scattering angles between 70 and about 100 degrees, both effective radius and optical depth show positive biases. These low scattering angles occur near the edge of the MODIS swath that is oriented toward sun. Since Aqua's local equator crossing time around 14:30, the sun is toward the west of the satellite; therefore the western edge of the scan is most strongly affected. Note, there are only relatively few observations at such low scattering
angles. This region also loosely corresponds to the region with low sunglint angle, discussed next.

- In terms of sunglint angle (Figure 4, right panels), dependencies can be observed at the low end of sunglint angle and, more strongly, at the high end of sunglint angles. The low end of sunglint angles occurs to the west of the swath, where the solar zenith angle equals the sensor zenith angle. High sunglint angles occur at the eastern scan edge. While some dependence on sunglint angle can be observed for low sunglint angles, the dominant variability
on this (western) side of the swath appears to lie in the scattering angle dependency, which is dominated toward the scan edge by variations in observer zenith angle. On the eastern side of the swath a similar picture emerges with an strong increase retrieved optical depth and, consequently, CDNC toward the (eastern) scan edge. For high sunglint angles the data density however is also relatively low.

To summarize, view angle dependencies are found both toward the eastern and western scan edge. Results here are largely
consistent with earlier results reported in (Maddux et al., 2010) and are likely caused by increasing field-of-view size and three-dimensional radiative transfer effects toward the edge of the scan. In addition, dependencies of retrievals on scattering angle are also found in the rainbow and backscattering of the phase function. These are likely caused by assumptions made in the retrieval process. From hereon forward we refer to all of these effects combined as 'retrieval artefacts'.

### 4.2. Impact Stratification and retrieval artefacts on climatology

Before we discuss the impact of retrieval artefacts on the climatology, we discuss the impact of the stratification on the CDNC climatology.

### 4.2.1. Impact of stratification

Figure 5 shows the mean relative difference 'Stratified' versus 'Non-Stratified' for the year 2008. Only 1x1 degree grid boxes are shown were all 12 months had valid values for both, 'Stratified' and 'Non-Stratified'. Differences are in general
positive with the exception of a narrow band in the tropical Pacific. Typically relative differences in mean climatology are smaller than 10 % with a few exceptions off the east coasts of Asia and North America. Figure 5 also shows the (2008) annual cycle for four selected areas. We selected areas where the difference between Stratified' and 'Non-Stratified' are particularly large. For example, in area X12 one can identify a secondary peak in the annual cycle in October that does not exist, if only stratified observations enter the climatology. The area BC1 shows an annual cycle offset by two months
between the two climatologies. The area R05 shows in general much lower CDNC and also a decreased annual cycle if the stratification criterion is not applied. In summary, we find that deviations in mean value between 'Stratified' and 'Non-



Stratified' are small but largely systematic globally. Further, the annual cycle of CDNC can be strongly affected, depending on the area observed.

### 4.2.2. Impact of retrieval artefacts

Figure 6 shows the impact of the above discussed retrieval artefacts on the CDNC climatology. The figure compares the 'stratified' results used in the final climatology with 'flagged results' in which sampling was only performed for observation geometries that did not show large variations in retrieved effective radius and optical depth (see Section 4.1 and Table 3). The additional constraints on observation geometry significantly reduce spatial coverage as can be seen in the upper panel of Figure 6. The relative difference in CDNC between 'flagged' and 'stratified' is less than 10 %. The annual cycles for the four selected areas also discussed under 4.2.1. are very similar between 'flagged' and 'stratified'. An exception to this is R03 where the 'flagged' results show an enhanced annual cycle over the 'stratified' results but does not show a shift in annual cycle. Other selected areas (not shown) substantiate these findings, i.e. the relative differences between 'flagged' and 'stratified' are small, the annual cycle appears to be not affected, but the magnitude of the annual cycle is somewhat muted in the 'stratified' cases. Clearly, while some of the retrieval artefacts propagate through into the climatology, their impact is somewhat reduced by averaging out strong the observation geometry dependencies seen in the Level 2 data (Section 4.1.) when the monthly 1x1 degree mean values are aggregated. Recall also that the retrieval artefacts cannot easily be fixed without potentially re-deriving new Level-2 cloud optical properties retrievals. While a restriction of the climatology to 'flagged' results would ameliorate some of the impact of the retrieval artefacts, it would at the same time severely limit the climatology spatially. Weighing these different aspects, we opted in favour of increased spatial coverage.

### 4.3. Assessment versus Vocals-Rex in-situ observations

To assess the robustness of the CDNC retrievals in this study, the results were compared to the in situ CDNC observations of (Painemal and Zuidema, 2011), hereafter referred to as PZ11. Twenty profile-averaged CDNC measurements were collected in 2008 during the VAMOS Ocean-Cloud-Atmosphere-Land Study Regional Experiment (VOCALS-Rex) coincident with Aqua and Terra overpasses. The PZ11 CDNC estimates were derived from the profile average of the onboard cloud droplet probe. Details on the sampling strategy are given in PZ11. Here we compare individual MODIS CDNC retrievals as well as areal averages of MODIS CDNC retrievals to the in-situ-derived CDNC values from PZ11. Nearest neighbour (NN) MODIS observation were found spatially corresponding to each profile's latitude and longitude as reported in PZ11. Temporally, either Aqua-MODIS or Terra-MODIS was used depending on the observation time reported in PZ11. In addition to the NNs also spatial averages of MODIS observations over 21x21 and 51x51 pixels were used, corresponding to areal extents of 0.2x0.2 degree and 0.5x0.5 degree respectively at nadir. The approach was performed independently for both stratified and non-stratified retrieval selections. For the non-stratified NN cases, the nearest pixel was generally within 2 km of the profile location. When considering stratified nearest neighbours, the distance tends to increase which may decrease coherence, but most stratified observations were within 10 km of the profiled cloud. Results are presented in Figure 7 and Table 1.



Regardless of grid-box size and whether or not the stratification criterion was applied, the comparisons between in-situ and remotely sensed CDNC show high correlations and biases in the order of at maximum 10% of the mean retrieved CDNC values. The non-stratified results show a generally good agreement with PZ11, with almost negligible bias and root mean square errors less than 25 cm$^{-3}$. For stratified results, magnitude of the RMSE and bias increase, but are generally also in agreement with PZ11. Various effects, including cloud top entrainment or the representativeness of the chosen condensation rate, can potentially cause the somewhat larger bias for the stratified cases. While the dataset compiled by PZ11 is extremely helpful, it is not globally representative. More in-situ comparisons, covering a wider variety of clouds and observation geometries are needed to further elucidate such effects on retrievals globally.

Results for the average 51x51 and 21x21 domain cases show results similar to the single pixel cases in terms of biases and RMSE values. However, for both 51x51 and 21x21 neighbourhoods, the mean uncertainty (size of error bars in Figure 7 and last two columns in Table 1) is reduced for stratified cases over the un-stratified cases. Recall that the mean uncertainty reported for these two neighbourhoods is the standard deviation of all valid CDNC retrievals in the respective averaging box. Effective radius stacking is likely decreasing the selection of pixels of inhomogeneous clouds or those subject to sub-pixel effects, which, as discussed in Section 3, can result in retrievals with a wide spread of unphysical effective radii.

Finally, we note that the uncertainties of the NN approach were calculated using Gaussian error propagation using the reported effective radius and optical thickness errors in the MODIS Collection 6 data following the approach in Bennartz (2007). One can see that the so-derived uncertainties are of about the same magnitude as the uncertainties derived from the standard deviations of the 21x21 and 51x51 boxes. This analysis holds true also for larger box sizes and a wider variety of cases (not shown). In the final CDNC climatology we therefore only report the standard deviation of the retrieved CDNC values within each 1x1 degree grid box as an uncertainty estimate.

## 5. Climatological results

### 5.1. Global overview

Figure 8 and Figure 9 show a global overview of the CDNC climatology. The upper two panels of Figure 8 show mean CDNC and relative uncertainty for the entire time period. The lowermost panel of Figure 8 shows the fraction of months with missing data. One can see that mean CDNC is typically high near the coasts, in particular downwind of the major continents, and lower over the remote oceans. In particular in the tropics certain areas exhibit very low data coverage. Areas in the northern Indian Ocean and the western Pacific exhibit no valid data at all. These areas are typically associated with convective regions where the ISBLC assumption frequently breaks down because either isolated cumuli or deep convection are observed. Near the fringe of these regions also the largest relative uncertainties are observed. Over the mid-latitude storm tracks as well as over the traditional stratocumulus areas west of the major continents, data coverage is higher and various regions show full coverage for all months. Relative uncertainty is in the order of 60% to 80% in the storm tracks and increases polewards. In the stratocumulus regions the relative uncertainty is about 30%. Recall that uncertainties are




calculated as the square-root of the mean of the daily 1x1 degree variances over the course of each month, thereby assuming uncertainties between days to be uncorrelated (see Section 2.3). The reported uncertainties are however very similar to the a-posteriori uncertainties derived from error propagation (see Section 4.3).

Figure 9 shows information on the annual cycle of CDNC. The data density in Figure 9 is lower than in Figure 8 because we

only included grid boxes where a full annual cycle was available. For some areas, in particular in the tropics, data was only available for certain months and these grid boxes were excluded from the annual cycle analysis. The upper panel of Figure 9 shows the magnitude of the annual cycle both color-coded and as isolines. These same isolines are over-plotted also over the other two panels of Figure 9. The strongest annual cycle is found off the coast of China and off the east coast of the North- and South America. Strong annual cycles are also found in the stratocumulus regions off the west coast of South America

and Australia and, to a lesser, extent, off the west coast of Africa. In all of these regions with strong annual cycle a large fraction of the CDNC variability is explained by simple cosine fits. In contrast, certain areas in the North Atlantic and off southern Africa show a more complicated behaviour although the annual variability there is lower (for examples, see Figure 10). The lowermost panel of Figure 9 shows the month of the peak CDNC. The month of peak CDNC varies greatly depending on region. Subsequently we discuss and highlight some of the regions in more detail.

## 5.2. Regional results

We present regional results for a variety of different areas that have been discussed in the literature (see Figure 8 for details). We present results in terms mean CDNC as well as trends in Table 2. In addition we show the time series as well as the average annual cycle for selected areas in Figure 10.

### 5.2.1. Remote Areas

We categorize as remote areas the areas R01, R06, K08, X20, and X21, the last three of which are also shown in Figure 10. The areas are sufficiently remote to be not strongly affected by anthropogenic changes in aerosol conditions to serve as a check for the temporal stability of the dataset. Indeed, for none of these areas trends in CDNC are larger than a $\pm3$ cm$^{-3}$ and none of trends are even remotely statistically significantly different from zero. We therefore conclude that the baseline of the climatology appears to be robust and the underlying MODIS observations do not exhibit any serious trends. Such trends

could for example have been caused by slight changes in absolute calibration or other sensor-specific issues.

### 5.2.2. Southeast Atlantic (off southern Africa) and south of Madagascar

The Atlantic region off southern Africa has been widely studied in the context of biomass burning and the potential impact of biomass burning aerosols on clouds via the first aerosol effect (Bennartz, 2007;Wilcox et al., 2009;Painemal et al., 2014). Notably, the annual cycle of CDNC peaking in July appears to be a persistent feature in CDNC climatologies (see Figure 9

and also area X12 in Figure 10) that is consistent with southern Africa biomass burning. Peak anthropogenic biomass burning does not coincide with the height of the dry season but does coincide with the middle of the burning season over the





northern parts of Angola, the Democratic Republic of Congo, and Zambia, where often Savannah fires are lit in July/August well before the peak of the dry season (Le Page et al., 2010). This picture is substantiated by individual fire counts based on geostationary satellite observations, which also shows peak fire activity around July/August with peak burning of about 6Tg per day in July (Roberts et al., 2009). In contrast, in the eastern part of South Africa and Mozambique as well as northern

Namibia, the middle of the fire season is shifted toward September. Depending on large-scale weather patterns biomass burning aerosol can be transported westwards over the southern Atlantic or eastwards over the southern Indian Ocean (see e.g. Sinha et al. (2003)). With the biomass burning season in south-eastern Africa and Madagascar peaking in September, one would expect to find the annual cycle in CDNC also peaking around that time and indeed the climatology does show exactly this (see Figure 9 and also area BC1 in Figure 10)

**5.2.3. East China Sea**

Various authors have studied the East China Sea in one form or another (see Figure 8, R05, BA5, BB1, X15 (shown in Figure 10), X16, X17). It is interesting to see that over the observation period of the current study all of these areas exhibit a negative trend, often statistically significant, in CDNC for the period 2003-2013, which is opposite the positive trend in CDNC reported for the area BB1 and the period 1982 - 2010 (Bennartz et al., 2011). In that earlier publication we have also

established a causal link between the Chinese $SO_2$ emissions and wintertime CDNC over the East China Sea. We speculate that decreases in Chinese $SO_2$ emissions over the last decade or so have partly reversed the effect seen in the original study Bennartz et al. (2011). This speculation is substantiated by other studies that report a slight decrease in Chinese $SO_2$ emissions from its peak value in the 2004 to 2006 time range, the decrease in $SO_2$ emissions largely caused by flue gas desulfurization technology installed in coal power plants (Lu et al., 2011). Satellite observations suggest an even larger

decrease in Chinese $SO_2$ emissions by 50 % for the recent time period 2012-2015 compared to the year 2005 (Krotkov et al., 2016).

**6. Conclusions**

The climatology described in this publication provides a number of incremental improvements over earlier climatologies published by us. Importantly, by making the climatology static and providing a DOI, we hope to be able to contribute to a

better traceable of results through different studies that might use this climatology. We believe this is a particularly significant issue, as design choices in the generation of the climatology, such as data screening, can have a major influence on scientific results. To this extent we have aimed at clarifying as accurately as possible the screening choices as well as any other decisions that went into the generation of this climatology. For example, we decided to screen out any observation where the three MODIS-derived effective radii are inconsistent with the underlying assumption of the ISBLC. While this

additional screening reduces data coverage, it at least partly guards against the misinterpretation as variability in CDNC of three-dimensional radiative transfer effects in broken clouds. We further show that neglecting this screening does not only



lead to moderate biases in the annually averaged CDNC but also to sometimes large shifts in phase and amplitude of the annual cycle of CDNC in various regions.

We found some remaining retrieval artefacts in the MODIS-retrieved effective radius and optical depth that propagate through into artefacts of the CDNC climatology as well. Our assessment of those effects on the climatology shows however

that effects clearly visible in the Level-2 data are averaged out to some degree in the aggregated climatology. We therefore did not in the final climatology include any additional screening for those effects. However, feel that it might be beneficial in future work to re-create retrievals working directly on the Level-1 reflectances that address some of the issues we have identified. Such retrievals could be specific to stratiform boundary layer clouds and might include more appropriate assumptions for example about the width of the droplet spectrum and the stratification of the cloud as well as a finer retrieval

grid or a direct calculation of the first one or two orders of scattering that corresponds to the actual observation geometry for each pixel. Another important issue not addressed here is the consistency between Terra-MODIS and Aqua-MODIS as well as transfer of the methods derived here to geostationary satellites, in particular the recently launched GOES-16 and Himawari satellites that both provide the necessary channel settings to derive three effective radii and in addition allow to capture the diurnal cycle of clouds with very high resolution.

Beyond these natural extensions of the current work, it will be important to devise new strategies to better derive CDNC from satellite using different types of satellite observations. Various approaches have been proposed and partially tested by other authors as outlined in the introduction. Only innovative approaches and new observational strategies will allow overcoming some of the more difficult issues that affect CDNC observations. Examples of such issues include overlying aerosol layers as well as the effects of entrainment on the vertical profile of clouds, where assumptions made in the simple

ISBLC could be replaced either by direct observations or by better assumptions that would make CDNC estimates more robust. Another important aspect is the need for more airborne in-situ observations of CDNC, collocated with satellite observations, also in regions outside of the traditional stratocumulus areas. While satellite observations and modelling studies based on high-resolution numerical models can help develop and improve CDNC estimates, only in-situ observations can be regarded as independent validation.

**Acknowledgements**

The authors are indebted to Rob Wood and Daniel Grosvenor for identifying and iteratively helping fix an error in the formulation of condensation rate in Bennartz (2007). The authors would further like to thank Veronica Ikeshoji-Orlati for her help with publishing the dataset via the Vanderbilt Library.





**Tables**

Table 1: Statistics of comparison between PZ11-reported airborne and satellite-based derived CDNC estimates, corresponding to the scatter plots shown in Figure 7. The columns marked 'N' refer to 'Non-stratified' case, whereas the columns marked 'S' refer to the 'Stratified' case. The reported uncertainty is the mean value of the error bars seen in Figure 7.

| Domain | RMSE [cm$^{-3}$] | | Bias [cm$^{-3}$] | | Correlation | | Uncertainty [cm$^{-3}$] | |
|---|---|---|---|---|---|---|---|---|
| | N | S | N | S | N | S | N | S |
| Nearest Neighbor | 22.1 | 29.7 | -2.7 | -17.3 | 0.97 | 0.97 | 36.7 | 35.6 |
| 21x21 | 21.6 | 31.7 | 3.8 | -16.9 | 0.97 | 0.96 | 28.3 | 19.8 |
| 51x51 | 23.1 | 37.4 | 1.1 | -18.7 | 0.96 | 0.94 | 42.1 | 31.5 |

Table 2: For all areas shown in Figure 8, this table lists the latitude/longitude boundaries, mean CDNC, standard deviation of CDNC, regression slope of CDNC against time, and the statistical significance of the slope. Rows highlighted green have a significance level higher than 95%. The regression slope was calculated on monthly CDNC anomalies. The monthly CDNC anomalies were calculated by subtracting from each individual month the 13-year mean value of that same month.

| Area | Left Lon | Right Lon | Lower Lat | Upper Lat | Mean | Stddev | Slope | Stat. Sig. |
|---|---|---|---|---|---|---|---|---|
| | deg E | deg E | deg N | deg N | cm$^{-3}$ | cm$^{-3}$ | cm$^{-3}$/10yr | % |
| X01 | -130 | -120 | 30 | 40 | 98.3 | 16.4 | -8.1 | 69.3 |
| X02 | -70 | -60 | 35 | 45 | 140.5 | 25.2 | -50.5 | 100.0 |
| X03 | -80 | -70 | 20 | 30 | 91.5 | 20.1 | -31.7 | 98.6 |
| X04 | -95 | -85 | 20 | 30 | 127.8 | 20.2 | -28.7 | 95.0 |
| X05 | -100 | -90 | 5 | 15 | 63.6 | 7.4 | 2.9 | 55.8 |
| X06 | -90 | -80 | 0 | 10 | 72.4 | 10.1 | -0.4 | 5.3 |
| X07 | -90 | -80 | -10 | 0 | 92.0 | 8.8 | -10.9 | 93.7 |
| X08 | -85 | -75 | -20 | -10 | 97.1 | 9.9 | -9.9 | 95.5 |
| X09 | -45 | -35 | -30 | -20 | 93.1 | 15.8 | -5.5 | 56.1 |
| X10 | -20 | -10 | 35 | 45 | 101.2 | 21.5 | -17.9 | 89.5 |
| X11 | -30 | -20 | 10 | 20 | 70.7 | 7.5 | -4.1 | 76.0 |
| X12 | 0 | 10 | -10 | 0 | 98.9 | 10.4 | -5.3 | 65.3 |
| X13 | 60 | 70 | 15 | 25 | 94.1 | 12.2 | 0.9 | 9.1 |
| X14 | 82 | 92 | 10 | 20 | 83.9 | 13.4 | -6.4 | 39.2 |
| X15 | 120 | 130 | 30 | 40 | 240.0 | 21.3 | -19.5 | 96.7 |
| X16 | 120 | 130 | 20 | 30 | 165.6 | 20.3 | -16.6 | 87.1 |
| X17 | 110 | 120 | 10 | 20 | 98.4 | 19.0 | -20.9 | 92.2 |





| | | | | | | | | |
|---|---|---|---|---|---|---|---|---|
| X18 | 120 | 130 | -10 | 0 | 67.1 | 11.1 | -2.5 | 18.1 |
| X19 | 90 | 100 | -15 | -5 | 48.0 | 8.4 | -1.4 | 23.8 |
| X20 | 0 | 120 | -50 | -40 | 74.2 | 5.0 | -3.1 | 74.0 |
| X21 | -170 | -90 | -50 | -40 | 64.7 | 4.7 | -1.8 | 51.3 |
| R01 | 88 | 103 | -40 | -30 | 53.8 | 5.0 | -3.1 | 76.7 |
| R02 | -140 | -115 | 15 | 35 | 85.0 | 10.1 | -8.6 | 86.8 |
| R03 | -10 | 15 | -25 | -5 | 89.5 | 9.8 | -6.8 | 87.1 |
| R04 | -90 | -70 | -28 | -8 | 90.3 | 8.1 | -6.9 | 93.9 |
| R05 | 105 | 150 | 10 | 40 | 136.9 | 11.1 | -15.6 | 99.0 |
| R06 | -175 | -135 | -35 | -20 | 45.4 | 4.2 | -1.7 | 59.3 |
| BA1 | -157 | -108 | 20 | 39 | 74.6 | 7.6 | -7.4 | 92.0 |
| BA2 | -43 | -9 | 14 | 43 | 74.6 | 8.0 | -8.8 | 95.4 |
| BA3 | -101 | -62 | -37 | -8 | 69.2 | 5.5 | -4.1 | 89.2 |
| BA4 | -20 | 18 | -34 | 0 | 76.3 | 6.8 | -4.2 | 84.0 |
| BA5 | 109 | 148 | 11 | 30 | 107.7 | 12.5 | -9.3 | 82.0 |
| BB6 | 120 | 132 | 18 | 34 | 169.6 | 17.0 | -17.1 | 96.0 |
| BC1 | 30 | 50 | -35 | -25 | 95.9 | 11.1 | 0.6 | 10.0 |
| K01 | -90 | -80 | -20 | -10 | 68.5 | 8.4 | -4.7 | 75.2 |
| K02 | 0 | 10 | -20 | -10 | 98.7 | 12.3 | -7.8 | 79.7 |
| K03 | -130 | -120 | 20 | 30 | 91.5 | 13.1 | -7.7 | 70.9 |
| K04 | 95 | 105 | -35 | -25 | 52.0 | 5.7 | -0.4 | 11.0 |
| K05 | -90 | -80 | 10 | 20 | 78.5 | 12.1 | -9.9 | 80.0 |
| K06 | 170 | 180 | 40 | 50 | 92.0 | 13.0 | -3.6 | 43.2 |
| K07 | -45 | -35 | 50 | 60 | 98.3 | 13.3 | -12.4 | 90.6 |
| K08 | -180 | 180 | -59 | -50 | 81.5 | 4.1 | 0.9 | 23.5 |
| K09 | 105 | 120 | 20 | 30 | 188.8 | 25.2 | -25.5 | 90.1 |



**Table 3: Terminology used to differentiate various screening tests performed on the Level-2 data.**

| Term used | Explanation |
|---|---|
| Stratified | All six screening criteria listed in Section 2.3 are used. This is the screening used for the final published climatology. |
| Non-stratified | Only screening criteria 1-5 listed in Section 2.3 are used. Therefore, in contrast to 'stratified' the effective radii are not required to increase monotonically between the 1.6 and 3.7 μm retrievals. |
| Flagged | In addition to screening criteria 1-6 in Section 2.3, the aggregation process excluded observations with a sun glint angle smaller than 35 degrees and scattering angles smaller than 95 degrees or larger than 165 degrees. |



**Figures**

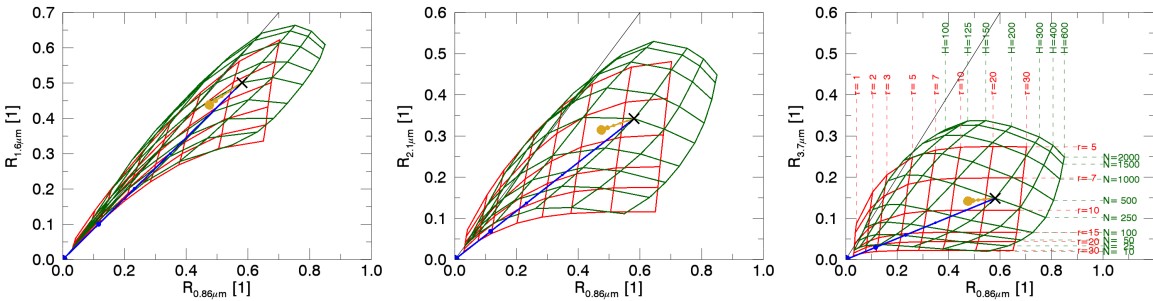

**Figure 1: Nakajima-King plots showing idealized relations between satellite-observed reflectances and retrieved cloud variables over a black surface for all three MODIS bands used for retrievals. The x-axis shows the reflectances at 0.86 μm, the y-axis, the reflectance at 1.6, 2.1, and 3.7 μm, respectively. The red mesh shows the dependency of retrieved optical depth ('τ', dimensionless, given in the rightmost panel) and retrieved effective radius ('r', in [μm]) on reflectance. The green mesh shows the dependency of retrieved CDNC ('N', in [cm⁻³]) and cloud geometrical thickness ('H', in [m]) on the reflectance. The thin black line is the one-to-one line for reference. The thick blue and golden lines indicate the impact of partly cloudy scenes (blue) and absorbing aerosol above the clouds (golden). More details are provided in the text.**





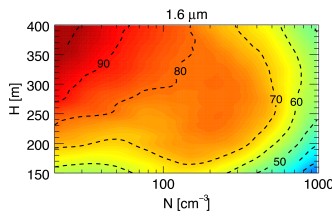 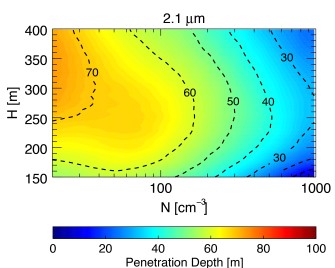 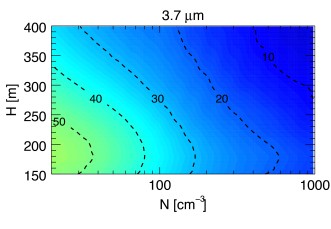

**Figure 2: Average penetration depth relevant for effective radius retrievals of CDNC in the ISBLC. For the purpose of this study, we define the penetration depths as the vertical distance below cloud top at which the retrieved effective radius equals the effective radius predicted by the ISBLC.**





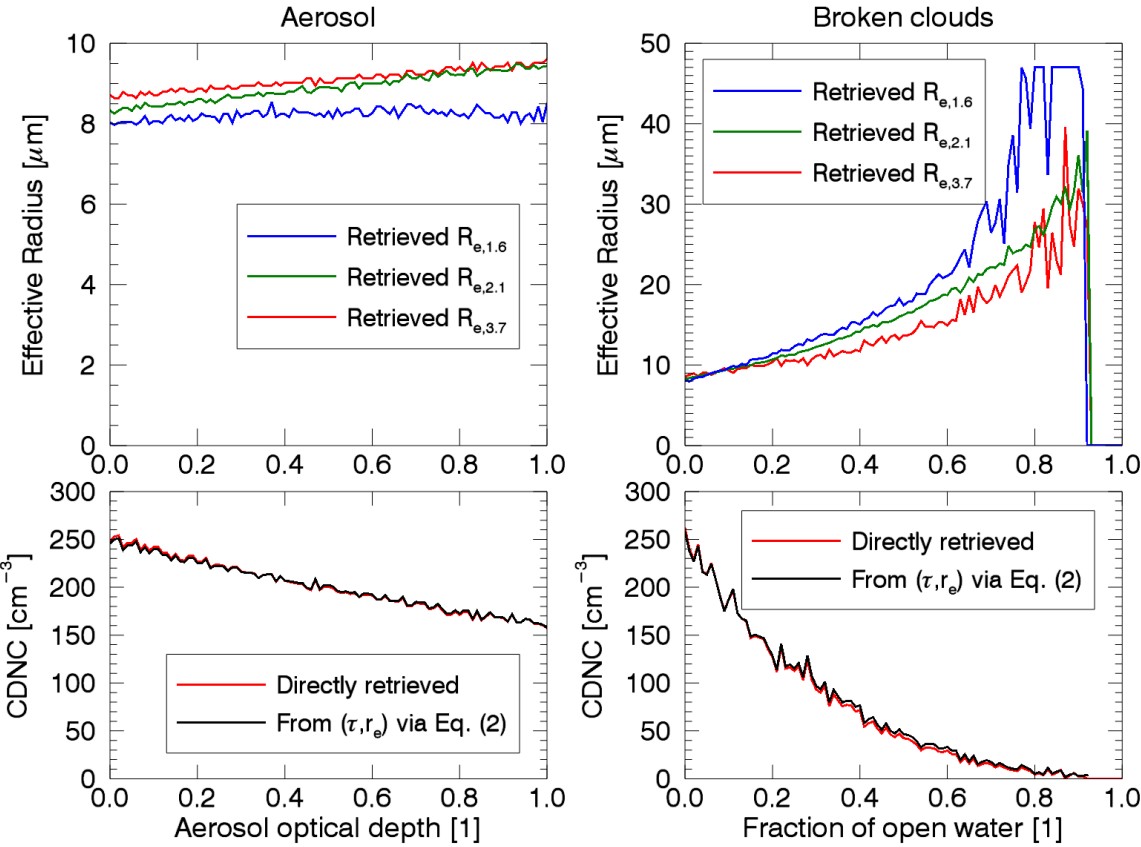

**Figure 3: Simulated retrieval for effective radius (upper panels) and CDNC (lower panels) for the aerosol-affected (left) and partly cloudy reflectances (right), respectively, shown in Figure 1. CDNC retrievals tagged 'directly retrieved' are performed using the green meshes in Figure 1, rightmost panel, whereas CDNC-retrievals marked as 'from Eq. (2)' are derived by first using the 3.7 μm optical depth and effective radius retrieval and subsequently converting it to CDNC using Eq. (2). This latter approach is similar to the actual way we retrieve CDNC from MODIS cloud optical properties.**




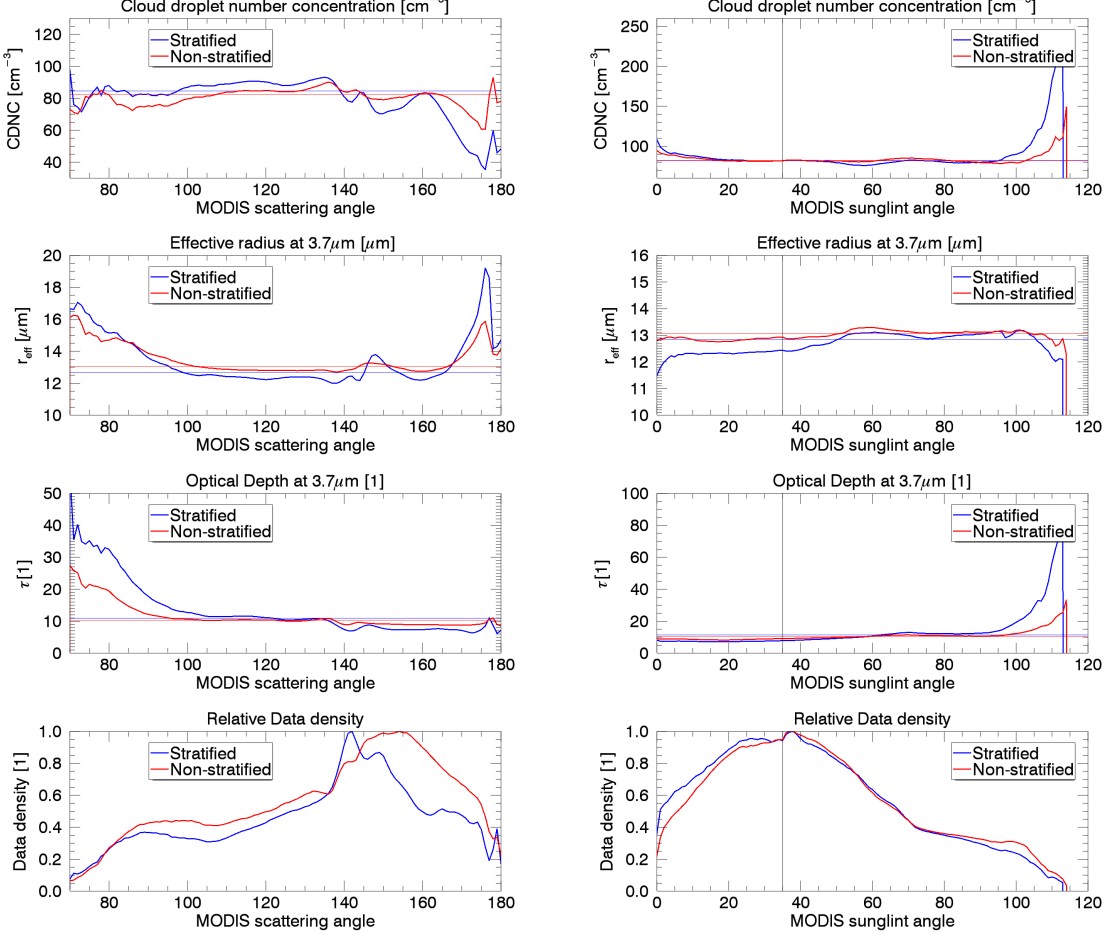

**Figure 4: Dependency of retrieved CDNC, effective radius and optical depth, all at 3.7 µm, on scattering angle (left panels) and sunglint angle (right panels) based on one year (2008) of Aqua MODIS observations. See Table 3 for terminology regarding 'Stratified' and 'Non-Stratified'. The vertical line in the right panels shows the sunglint angle below which the MODIS cloud masks flags a sunglint contamination. The horizontal red and blue lines are the average retrieved values over the entire dataset.**





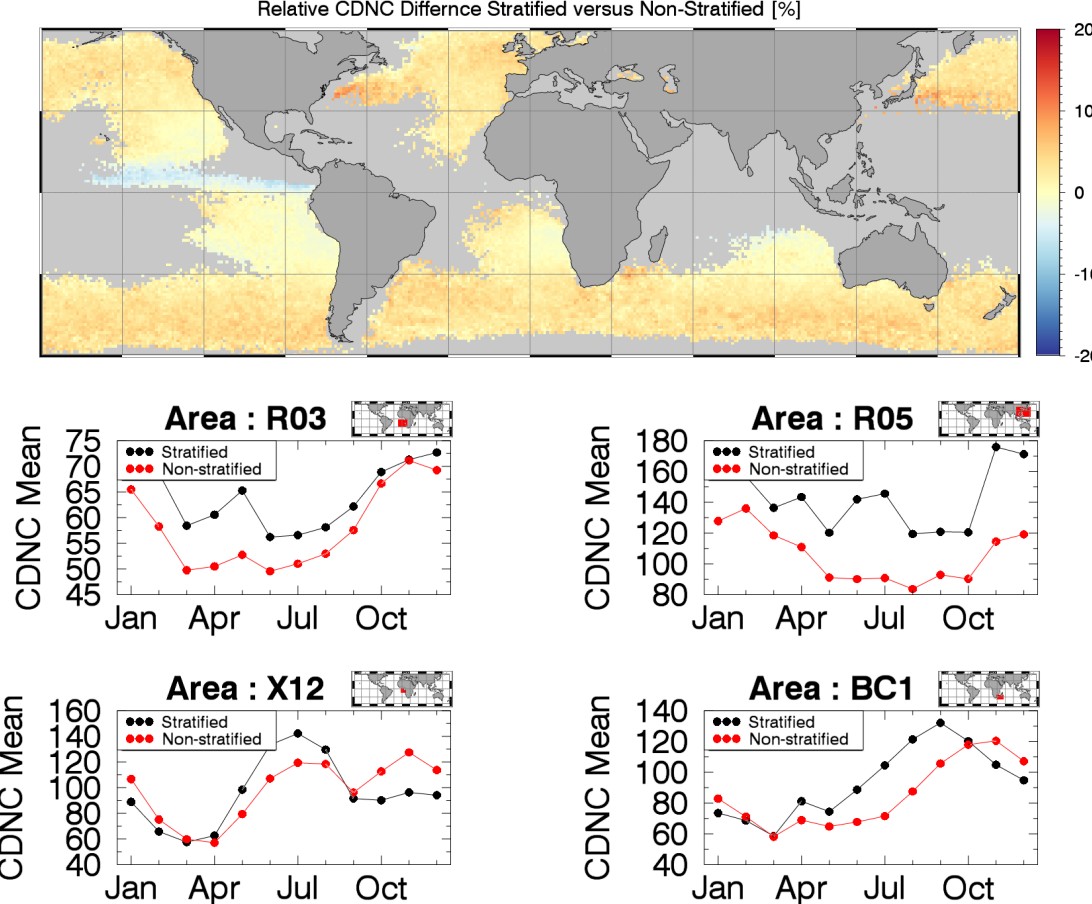

**Figure 5: Comparison of results for 2008 for 'Stratified' versus 'Non-stratified' climatologies; see Table 3 for terminology. The upper panel shows the relative difference in 2008 mean CDNC between stratified and non-stratified. The four lower panels show the annual cycle of CDNC for four selected regions where differences between stratified and non-stratified were particularly strong. For details on the selected regions, see also Table 2.**





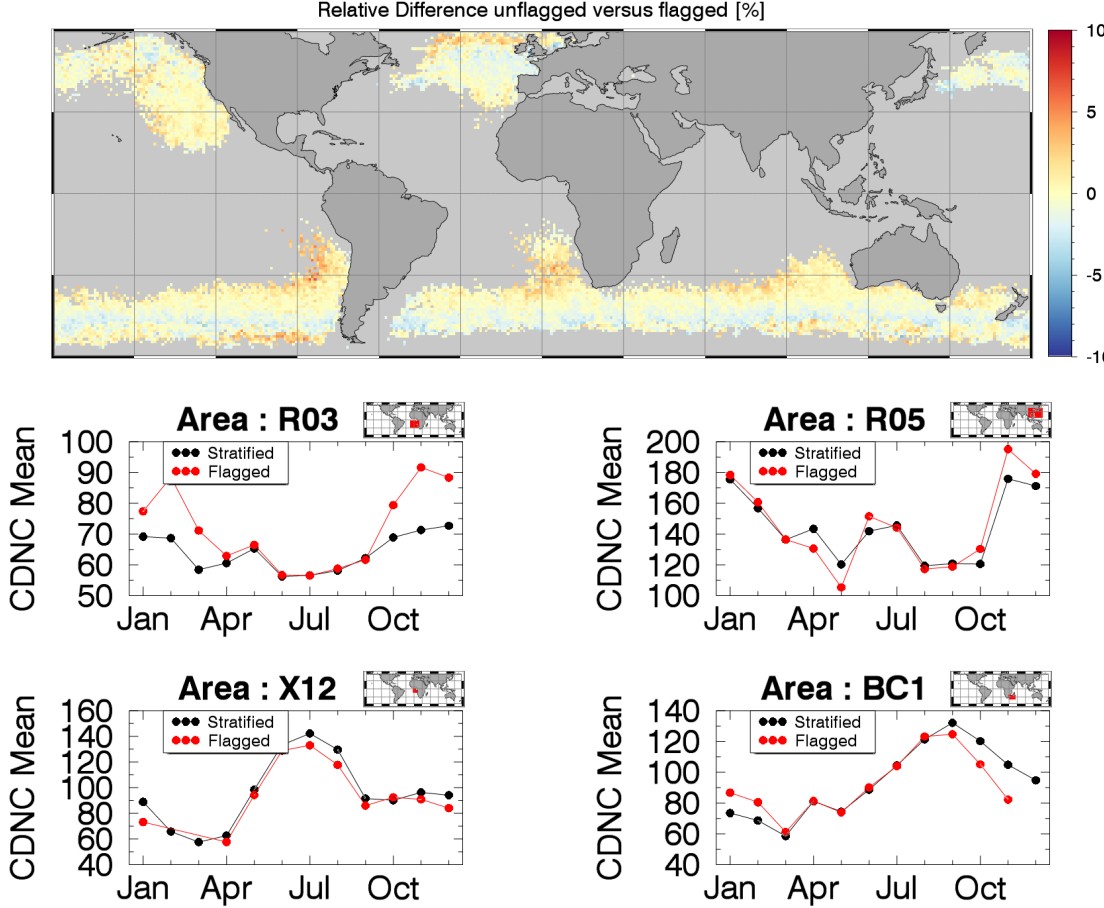

**Figure 6: Same as Figure 5 but for 'Stratified' versus 'Flagged'; see Table 3 for terminology. For details on the selected regions, see also Table 2.**





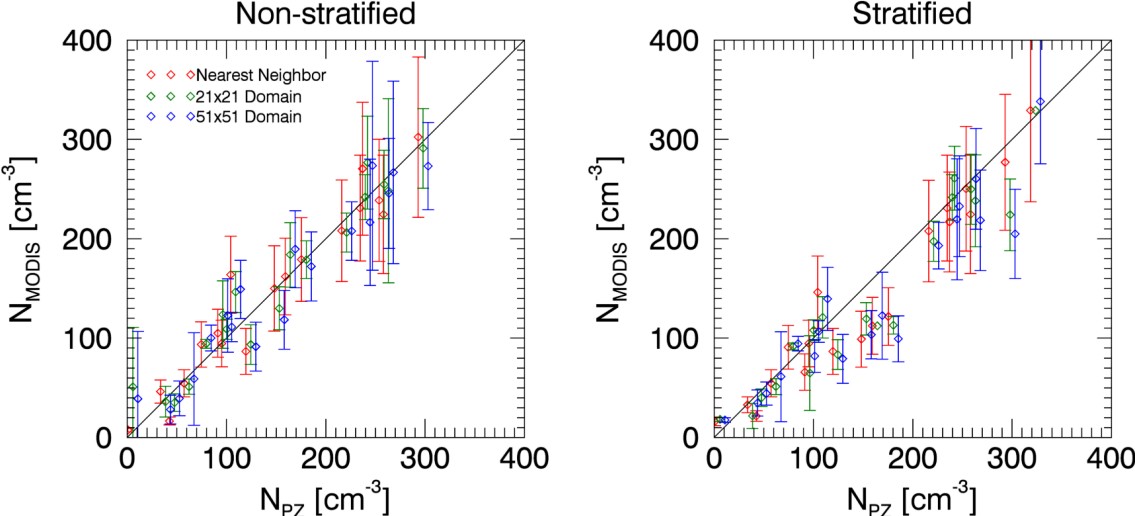

**Figure 7: Scatter plot between PZ11-reported airborne and satellite-based derived CDNC estimates, corresponding to the statistics shown in Table 1. Uncertainty for nearest neighbor is estimated using Gaussian error propagation of uncertainties of effective radius and optical depth reported in MODIS C6. Uncertainties for the 21x21 and 51x51 domains are estimates as the standard deviation of all valid CDNC retrievals within the domains. Note, that for each data point the x-axis values of the NN, 21x21, and 51,x51 are slightly offset around the (green) center value, in order to be able to better visually discriminate the error bars. See Table 3 for terminology regarding 'Stratified' and 'Non-Stratified'.**





**Figure 8:** The upper panel shows the mean CDNC over the period 2003-2013. The middle panel shows the mean relative uncertainty of the CDNC estimates and the lower shows the fraction of months with missing data in the climatology. Areas BA1-BA7 are defined in Bennartz (2007), BB1 in Bennartz et al. (2011), BC1 in this publication, K01-K09 in Klein and Hartmann (1993), R01-R06 in Rausch et al. (2010), and X01-X21 in Zhao et al. (2016). The corresponding latitude/longitude boundaries are also listed in Table 2.





**Figure 9: The upper plot shows the relative magnitude of the 13-year mean annual cycle of CDNC. The middle plot shows the variance explained by a simple cosine-fit of the 13-year mean annual cycle. The lower plot shows the months of the occurrence of the maximum CDNC. The isolines in all three plots show the relative magnitude of the CDNC annual cycle (same as upper plot).**





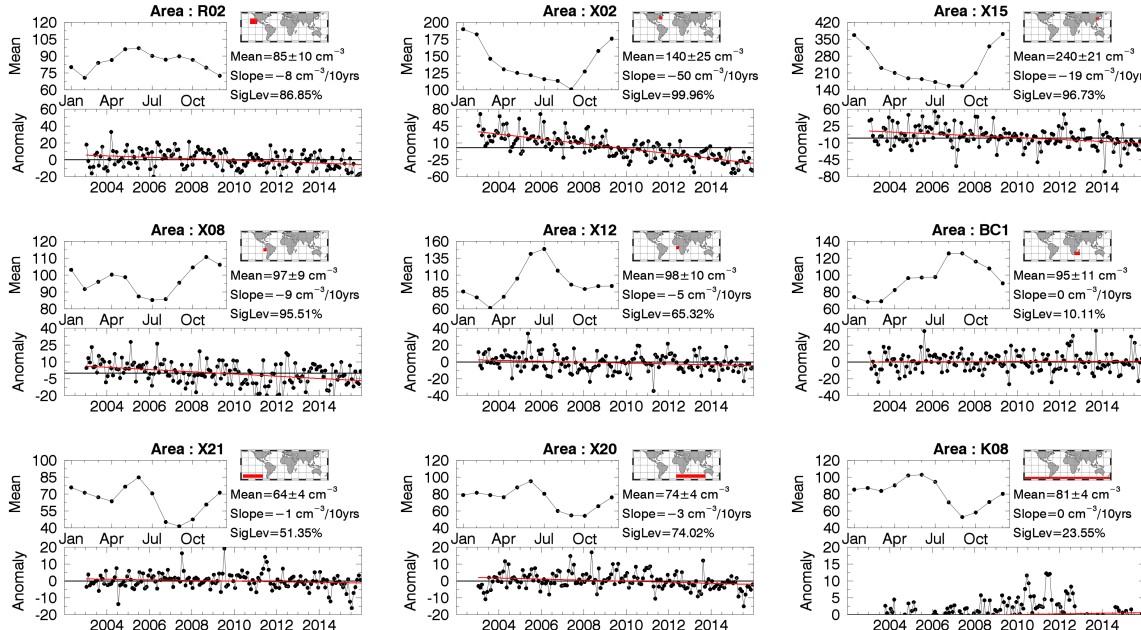

**Figure 10: Mean CDNC annual cycle, monthly CDNC anomalies, and statistics for selected areas as shown in Figure 8 and summarized in Table 2. The red curve in the anomaly plots is the regression line.**





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
