# Peer review of "Global and regional estimates of warm cloud droplet number concentration based on 13 years of AQUA-MODIS observations"

_Atmospheric Chemistry and Physics, 2016_

## Short Comment (SC1) · 17 Jan 2017

There is a strange sentence in the abstract:

"All values are in-cloud values, that is, if the grid box is covered to 10% with clouds, then the reported CDNC value will be valid for the cloudy part of the grid-box. "

As it is written, it means to me:

"If the grid box cloud cover equals exactly 10%, the reported CDNC values are valid for the cloudy part of the grid-box, otherwise they are undefined".

Does it mean

(A) that you use 10% as a minimum required cloud-fraction somewhere in the algorithm (Could not find that in section 2.3 where the gridding algorithm is explained)

(B) That you only report in-cloud values all the time, but use 10% as an example of a typical cloud-cover value in that sentence in the abstract? In that case, the sentence could be

"All values are in-cloud values. The reported CDNC-values are valid only for the part of the grid-box which is covered by clouds"

---

## Referee Comment (RC1) · Anonymous Referee #1 · 5 Feb 2017

Bennartz and Rausch update and discuss a very useful satellite dataset of liquid-cloud droplet number concentration over oceans. The paper is pertinent to Atmos. Chem. Phys. and very well written. It is very useful that the authors make available their dataset.

I have two major and a couple of specific suggestions that I think the authors should address before publication in Atmos. Chem. Phys.

Major comments

1. The authors introduce the notion of the idealized stratiform boundary layer cloud model. But they leave it unclear what the difference is to the previous adiabatic assumption. Do they include a profile of sub-adiabaticity as Boers et al. (2006)? Is CDNC in this model not vertically uniform and/or the liquid water content not linearly increasing with height? What is the assumption on sub-adiabaticity?

2. The uncertainty analysis is highly superficial. Fundamentally, the authors just write that in a single case (VOCALS, 20 flights) the "error" as propagated from the MODIS-retrieved reff and tau uncertainty assessed by Platnick et al. (2015) is similar in magnitude as the spatial variability at the scale considered for these cases (up to 51x51 km2). As such, it is highly astonishing that the authors take the spatial variability at face value as the uncertainty for any other cloud regime as well. The result of course is foreseeable: the variability is small for stratiform, and large for broken clouds. Although it is not unlikely that the actual error behaves like this, it cannot be concluded from the analysis by Bennartz and Rausch. I suggest the authors either perform a rigorous uncertainty analysis, or else abstain from calling the variability "uncertainty", but actually call it, e.g., "sub-scale variability".

Specific comments

Title: clarify this is only over ice-free oceans (maybe in the abstract is also sufficient)

p2 I 5: Studies preceding the Bennartz (2007) satellite climatology should be acknowledged, such as Han et al. Geophys. Res. Lett. 1998 (AVHRR) and Quaas et al. Atmos. Chem. Phys. 2006 (MODIS).

P3 I14: what is "maximum" adiabatic value here?

P3 I16: probably something like "...a growing body of work has been devoted to understanding..."?

P3 I20: what is the distinction of CDNC at cloud base and the one "observed"? This should be clarified, since for the assumption of vertically constant CDNC, there seems to be no difference.

P4 I6: should read km\$^2\$

**ACPD**
p4 l8: specify what type of observations

p4 I18: \citep[e.g.,][]{brenguier} and later citet{bennartz}

p5 I3: it would be good to give the reader some insight about the aircraft-observed range in k. E.g. Freud and Rosenfeld (J. Geophys. Res. 2012) find k = 0.93, while values from Martin et al. 1994 and Pawlowska and Brenguier (Tellus 2003) go down to 0.67. This introduces an uncertainty of about 20%.

p5 l16: "for the 13 years"

p6 I1: better clarify: "liquid water cloud"

p6 l28: some more detail is required how this series of reflectances was constructed. Which thermodynamic profiles, cloud-base heights and cloud-base updraft speeds / CCN concentrations were sampled?

P7 I21: "horizontal and vertical" ?

P7 I26: black cross? And CDNC = 500 cm-3 / H = 175 m?

P8 I5: It would be useful if the authors discussed how often the MODIS algorithm fails to diagnose partly cloudy pixels, i.e. how important this problem remains after condition (4) on p6 is enforced.

P8 I12: correct color and numbers

p8 I13: "an optical depth"

p8 I15: the assumed single scattering albedo should be reported p8 I16: why "not shown"? It is represented in Fig. 1 as well, as far as I understand.

P9 I6: "angle increases" or "angles increase"

p9 I16: the screening criteria should be explained (reference to Tab. 3)

P9 I20: "from"
p9 l22: "in a manner that"

p10 I2: "biases" - it is not a priori clear that these are biases, or do I miss a point?

P10 I15: citet

p11 I2: The authors should report how many datapoints are sorted out by the stratification criterion.

P11 I7: From Fig. 4, it seems the problem of reduction in amount of data is mainly due to the sunglint angle >  $35^{\circ}$  criterion. However, compared to the scattering angle criterion, this one seems to be of minor importance. I suggest to split the two issues and investigate them separately.

P12 I20: This is a very brief uncertainty quantification discussion. The notion that spatial variability could fully represent the error seems implausible.

P13 l22: the trend should have a per time unit.

P13 I28: aerosol indirect effect? Or rather "radiative forcing due to aerosol-cloud interactions"?

P14 I25: traceability

p20, Fig. 2: It would be desirable to add smaller CDNC to the plot, since CDNC < 20 cm-3 are frequently retrieved.

P21, Fig. 3: The caption should explain the label "fraction of open water": is this 1 - cloud fraction? Also the scale is unclear, is this at the 1x1 km2 scale?

P22, Fig. 4: Are the curves averaged over the year and globe for given scatter-ing/sunglint angle?

P23 Fig. 5: title top panel "Difference"

p26 l8: this is the central result of the climatology, so it deserves more attention. I suggest to move the boxes from the top panel to the (less important) bottom panel.

**ACPD**
Rather than reporting the number of missing months, I suggest to report the fraction of missing days in each  $1^{\circ}x1^{\circ}$  grid box. For the climatology (top panel) a color should be chosen that better allows to distinguish CDNC at lower concentrations, possibly a non-linear color scale would be very helpful in this regard.

p29 references: journal names should be abbreviated. Some titles are in upper cases.

---

## Referee Comment (RC2) · Anonymous Referee #2 · 7 Feb 2017

**Review of Bennartz and Rausch – "Global and regional estimates of warm cloud droplet number concentration based on 13 years of AQUA-MODIS observations"**

This paper describes a climatology of monthly mean cloud droplet concentrations (CDNCs) using the MODIS-Aqua satellite instrument. Some of the potential retrieval issues that can affect CDNC retrievals are discussed along with some examples of maps and seasonal cycles from the final climatology. This data set is likely to be a very useful commodity to many researchers and I commend the authors for making it available online. Thus I recommend the publication of the paper once certain issues have been addressed and additional discussion points added. The major issues are summarized next with more specific comments and typos below that.

**Major issues**

One key point on which I disagree with the authors is their restriction to situations where re,3.7 > re,2.1 > re,1.6. Painemal and Zuidema (2011) showed that this criteria was often violated for MODIS retrievals of stratocumulus (generally re1.6, re2.1 and re3.7 were very close) and yet the aircraft observed LWC was found to increase with height. This is likely due to other errors in the retrieval of reff that affect the different MODIS channels in different ways. E.g. sub-pixel heterogeneity can increase re2.1 more than re3.7 even in relatively homogeneous stratocumulus (Zhang, 2012), and Grosvenor (2014) showed that resolved heterogeneity is likely to affect the re3.7 more than re2.1. Also, King (2012) show that the information content from the different MODIS channels is not enough to be able to retrieve vertical height variation in reff and so it seems to me that this restriction might be causing some spurious filtering of the data. It is also not clear from Fig. 3 that re3.7 > re2.1 would select the cloud covered part of the curve as stated since re3.7 and re2.1 are very close, as was also seen in the Painemal and Zuidema (2012) aircraft observations. Differences between the stratified and non-stratified CDNC values are reported in Section 4.2.1, but can we be sure which is more realistic? I think that there should be more discussion about and acknowledgement of these issues.

Consideration of scattering angle effects and the effect of sunglint is made. However, the applied filtering does not take into account the likelihood of retrieval biases at high solar zenith angles. A little more detail should be given on this issue along with consideration on how this bias will affect your data set. For example, this could potentially cause large-scale biases in the winter months at high latitudes.

It can't be said for sure whether some of the scattering angle and sunglint effects in Section 4.1 are due to variations in region – certain regions or seasons may be sampled more often at certain scattering angles and with certain sunglint angle. And there can be strong natural regional changes in reff, tau, CDNC, etc. Is there a way to rule this out? E.g. can you look at the differences from within the same swath relative to the swath mean and then average these, so that the locations of samples are roughly the same? This is possible for sensor angle (as in Maddux, 2010) and sunglint angle, but I'm not sure about scattering angle. There is also no direct examination of the effect of at sensor angle. This would be

useful, or at least you could directly refer to this from the results of Maddux (2010)? Also, the swath edges may also sample a different range of scattering angles than elsewhere in the swath, which may also cause artifacts. Have you looked into this?

RE "uncertainties" – using the variability of the daily CDNC values means that a lot of what is termed "uncertainty" will actually be natural variation in CDNC, which, even with perfect retrievals, would not be expected to be constant throughout a cloud field, nor over the course of a month, nor over the climatology. I understand that given the fact that the uncertainties are not well characterized it is hard to separate the real variability from the noise introduced by retrieval errors. However, perhaps the quoted term should just be renamed "standard deviation" with it being made clear that some of this will be actual variability and some uncertainties in tau and reff and I think it would be useful if these were provided to get a sense of how much they contribute. However, it should also be made clear that these would not account for the entirety of the uncertainty since there will be a lot of error introduced by the forward model plane parallel independent pixel approximation that will not be captured.

Fig. 5 - it looks like there is very little data in box X12 – the numbers of samples going into each box should be quoted for this figure. There is also overlap between box R03 and X12, yet box X12 shows quite different results, perhaps indicating that the statistics for this box are too poor to be robust?

**Specific comments**

Abstract – "Resulting CDNC uncertainties for the climatology are in the order of 30% in the stratocumulus regions and 60% to 80% elsewhere" – as discussed in the other comments is it appropriate to call this "uncertainty". A more careful description is needed here.

**p.2, L27 – "1. The cloud is assumed to be horizontally homogeneous."** – This seems a little vague. It would be good to mention over what scales the cloud needs to be horizontally homogeneous and why. E.g. 1km, 1x1 degree or both? Or larger even? Perhaps it would be worth mentioning the application of the independent pixel approximation for reff and optical depth retrievals such that inhomogeneity over scales larger than the 1km pixel size is likely to violate this assumption with the scale being set by the degree to which net horizontal photon transport occurs (the scale of any shadows/bright spots, etc., or rather deviations from the plane-parallel reflectances, as defined in Marshak, 2006). This could be separated from the requirement of homogeneity within the 1km pixel to satisfy the plane-parallel retrieval assumption. In terms of the CDNC derivation itself only the sub-pixel homogeneity is explicitly required once we have the correct 1km reff and optical depth since 1km resolution CDNC values can be calculated.

p.3, L16 – "The true three-dimensional variability of clouds poses significant challenges to any remote sensing algorithm and a growing body work has been devoted toward understand the impact of this variability on remote sensing estimates of CDNC." – these works should be cited here, or else you should refer to them in another section in the paper.

p.3, L19 – "as ultimately one would be interested in the number of cloud droplets activated at cloud base and not the number of cloud droplets observed" – I think this would depend on the application.
Some studies may be interested in how the cloud top CDNC might change due to lateral mixing, evaporation, etc., or removal of CDNC by precipitation and not necessarily just the cloud base CDNC. I can see that the cloud base CDNC would be of interest for comparing to model processes, but I think that the statement here generalizes too much.

**p. 4, Eqns. 2 & 4** – You should describe how cw (condensation rate) was calculated – did you use the MODIS cloud top temperature? What pressure did you use? Did you use the full adiabatic value, or 80% of this as mentioned on p.3. An issue with the calculation used in Bennartz (2007) is mentioned in the acknowledgments - it would be good to add more detail in the main part of the paper about this issue along with a reference for the correct formula – e.g. Ahmad (2013). A discussion of the compensation of errors in k, fraction of adiabatic condensation rate, reff errors found by Painemal and Zuidema (2012) should also be included.

**p.6 – "3. The cloud mask had to indicate the observation to be cloudy but not over ice or land."** – does this include the filtering of sea-ice covered regions? How robust is the detection of sea-ice? This is important since this is likely to lead to a poor retrieval.

**p.6** – "6. Observations were only considered, if the three MODIS-retrieved effective radii stacked up as re,3.7 > re,2.1 > re,1.6 ,as observations violating this criterion will also violate the key assumption of a vertically increasing LWC in the ISBLC." – this is a key point on which I disagree with the authors – Painemal and Zuidema (2011) showed that this criteria was often violated for MODIS retrievals of stratocumulus (generally re1.6, re2.1 and re3.7 were very close) and yet the aircraft observed LWC was found to increase with height. This is likely due to other errors in the retrieval of reff that affect the different MODIS channels in different ways. E.g. sub-pixel heterogeneity can increase re2.1 more than re3.7 even in relatively homogeneous stratocumulus (Zhang, 2012), and Grosvenor (2014) showed that resolved heterogeneity is likely to affect the re3.7 more than re2.1. Also, King (2012) show that the information content from the different MODIS channels is not enough to be able to retrieve vertical height variation in reff and so it seems to me that this restriction might be causing some spurious filtering of the data.

**p.6, L24 – "a solar zenith angle of 56 degrees. The resulting scattering angle of 124 degrees"** – it would be good to define what you mean by the scattering angle. I.e. that it is the angle between the sun and the satellite within the plane of the sun and the satellite and that 180 degrees is backscatter.

p.7, L23 – "Typically, most of these issues affect the effective radius at 1.6  $\mu$ m and 2.1  $\mu$ m more strongly than the effective radius at 3.7  $\mu$ m (Zhang et al., 2012b)." – this was true for the sub-pixel optical depth heterogeneity effect, but are there other examples? Grosvenor (2014) showed that the 3.7  $\mu$ m channel is likely to be more strongly affected by resolved 3D radiative effects, so this is not always the case. Although the latter effect is more likely at higher solar zenith angles.

**p.7, L33** – "observations (i.e. if cloud thickness and liquid water path varies within the field-of-view)." – Strictly, Zhang (2012) talk about the effect of having sub-pixel heterogeneity in optical depth (which cloud arise through combinations of LWC, CDNC, etc.) changes, rather than cloud thickness and LWP.

**p. 8, L 4 – "so that in the example shown in Figure 3 (right panels) the largest part of the retrievals would be (correctly) rejected."** – it is not clear from Fig. 3 that re3.7 > re2.1 would select the cloud covered part of the curve since re3.7 and re2.1 are very close, as was also seen in the Painemal and Zuidema (2012) aircraft observations. Why not just filter based on the cloud fraction?

**p.9, L4 – "Grosvenor and Wood (2014) address issues the dependency of cloud microphysical retrievals on solar zenith angle."** – However, it seems that the filtering applied does not take into account the likelihood of retrieval biases at high solar zenith angles and this issue is not brought up again. A little more detail should be given on this issue along with consideration on how this bias will affect your data set.

**p.9, L14 & Fig. 4** – sunglint angle is not defined. I'm guessing that you mean the absolute difference between the sensor zenith angle of a given pixel and the sensor zenith angle of the middle of the sunglint band within the swath, but it is not obvious.

Section 4.1 – It can't be said for sure whether some of the effects seen are not due to variations in region – certain regions or seasons may be sampled more often at certain scattering angles and with certain sunglint angle. And there can be strong natural regional changes in reff, tau, CDNC, etc. Is there a way to rule this out? E.g. can you look at the differences from within the same swath relative to the swath mean and then average these, so that the locations of samples are roughly the same? This is possible for sensor angle (as in Maddux, 2010) and sunglint angle, but I'm not sure about scattering angle.

**p.10**, **L10** – presumably high sunglint angles (if my definition of this is correct) will be associated with high sensor angles since they will have to be near the edges of swaths. So, this could be contribute to the artifact. Why not also look at sensor angle directly? Or at least directly refer to it from the results of Maddux (2010)? However, the above interpretation seem inconsistent with Maddux (2010). They showed that the swath edge liquid optical depth values were lower and the re higher than those near nadir. Are the results consistent with Maddux (2010) as stated at line 15?

Also, the swath edges may also sample a different range of scattering angles than elsewhere in the swath, which may also cause artifacts. Have you looked into this?

**Section 4.2.1** – differences between the stratified and non-stratified CDNC values are reported, but can we be sure which is more realistic? Also, it looks like there is very little data in box X12 – the numbers of samples going into each box should be quoted for Fig. 5. There is also overlap between box R03 and X12, yet box X12 shows quite different results, perhaps indicating that the statistics for this box are too poor to be robust?

**Fig. 6** – the title uses the term "unflagged", but "stratified" would be more consistent with the rest of the paper.

Also, the frequent positive values seem inconsistent with that expected from Fig. 4 where the main effect appears to be a decrease in CDNC at high scattering angles. Or is the increase at low sunglint angles dominating for these regions? Why not also exclude the very high sunglint angles for the flagged pixels?

**p.12**, L 5 – "PZ11. Various effects, including cloud top entrainment or the representativeness of the chosen condensation rate, can potentially cause the somewhat larger bias for the stratified cases." – It's hard to see how this might be the case? Can you explain? Could it also be that the stratified cases might be further from the aircraft observation?

p.12, L10 – "However, for both 51x51 and 21x21 neighbourhoods, the mean uncertainty (size of error bars in Figure 7 and last two columns in Table 1) is reduced for stratified cases over the un-stratified cases. ... Effective radius stacking is likely decreasing the selection of pixels of inhomogeneous clouds or those subject to sub-pixel effects, which, as discussed in Section 3, can result in retrievals with a wide spread of unphysical effective radii." - Could this also be due to there being fewer samples in each region due to being more selective (since this would reduce the std. deviation)? Or were all boxes chosen to have all possible pixels present?

**p.13, L3** – RE "uncertainties" – using the variability of the daily CDNC values means that a lot of what is termed "uncertainty" will actually be natural variation in CDNC, which, even with perfect retrievals, would not be expected to be constant throughout a cloud field, nor over the course of a month, nor over the climatology. I understand that given the fact that the uncertainties are not well characterized it is hard to separate the real variability from the noise introduced by retrieval errors. However, perhaps the quoted term should just be renamed "standard deviation" with it being made clear that some of this will be actual variability and some uncertainty. Also, since you can calculate the propagated CDNC uncertainty from the pixel level uncertainties in tau and reff and I think it would be useful if these were provided to get a sense of how much they contribute. However, it should also be made clear that these would not account for the entirety of the uncertainty since there will be a lot of error introduced by the forward model plane parallel independent pixel approximation that will not be captured.

p. 13 & Fig. 9 – "magnitude of the annual cycle" – can you state somewhere what you actually mean by this? Is it the peak-to-peak amplitude of the annual cycle (i.e. max monthly mean minus min monthly mean), or is it calculated based on the cosine fits? "Amplitude" would be a better word than "magnitude" since it relates more directly to the definition.

Fig. 10 – the bottom right panel needs the y-axis range changing to show the negative values.

**p. 14, L31** – "We further show that neglecting this screening does not only lead to moderate biases in the annually averaged CDNC" – Are these biases or just differences? How do we know which one is correct due to the issues raised earlier in this review?

**p.15, L32** – "We found some remaining retrieval artefacts in the MODIS-retrieved effective radius and optical depth that propagate through into artefacts of the CDNC climatology as well." – it would be useful to mention the issues that you are talking about here.

**p.15**, L6 – "However, feel that it might be beneficial in future work to re-create retrievals working directly on the Level-1 reflectances that address some of the issues we have identified." – although, perhaps it should be mentioned that it may not be possible to resolve many of these issues even if reprocessing Level-1 data – e.g. it would be hard to correct biases for 3D radiative effects, sunglint effects, etc.

**Tables** – it would make sense to alter the order of these to align with the order of mention in the paper – i.e. first Table 3, then Table 2 and then Table 1.

**Typos**

p.9, L4 – "Grosvenor and Wood (2014) address issues the dependency of" – remove "issues".

p.9, L20 – "Fro Figure 4"

p.9, L22 – "vary in a manner the roughly resembles"

p.10, L3 – "Since Aqua's local equator crossing time around 14:30,"

p.10, L13 - "with an strong increase retrieved optical depth" -> "with a strong increase in retrieved optical depth"

p. 10, L19 – "4.2. Impact Stratification and retrieval artefacts on climatology" -> "Impact of Stratification..."

p.10, L24 - "boxes are shown were all 12 months"

p.12, L2 – "in the order of" -> "on the order of"

p.12, L4 – "For stratified results, magnitude of the RMSE and bias increase" -> "For the stratified results, the magnitude ..."

p.14, L25 – "a better traceable of results" -> "a better traceability of results"

p.15, L6 – "However, feel that"

p.20, L20 - "cover have discussed" -> "have been discussed"

**References (only ones not cited in the paper under review are listed)**

Ahmad, I., Mielonen, T., Portin, H. J., Arola, A., Grosvenor, D. P., Mikkonen, S., Leskinen, A., Komppula, M., Lehtinen, K. E. J., Laaksonen, A., and Romakkaniemi, S.: Long term measurements of cloud droplet concentrations and aerosol-cloud interactions in boreal boundary layer clouds, Tellus B, 65, 20138, doi:10.3402/tellusb.v65i0.20138, 2013.

Marshak, A., Platnick, S., Varnai, T., Wen, G., and Cahalan, R.: Impact of three-dimensional radiative effects on satellite retrievals of cloud droplet sizes, J. Geophys. Res.-Atmos., 111, D09207, doi:10.1029/2005JD006686, 2006.

King, N. J. and Vaughan, G.: Using passive remote sensing to retrieve the vertical variation of cloud droplet size in marine stratocumulus: an assessment of information content and the potential for improved retrievals from hyperspectral measurements, J. Geophys. Res.-Atmos., 117, D15206, doi:10.1029/2012JD017896, 2012.

---

## Author Comment (AC1) · 22 May 2017

Thanks for this comment. We have changed the abstract accordingly.

---

## Author Comment (AC3) · 22 May 2017

The comment was uploaded in the form of a supplement:
http://www.atmos-chem-phys-discuss.net/acp-2016-1130/acp-2016-1130-AC3-supplement.pdf

---

## Author Response (AR1)

Response to Anonymous reviewer 1

We thank both reviewers for their in-depth reviews and their thoughtful comments. In response to the reviewers' main criticism we have added another section and figure dealing with uncertainty estimates and outlining our argument for why we think the standard deviation is the better uncertainty estimate. Below please find our point-by point response to the reviewer's concerns.

**Major comments**

**1. The authors introduce the notion of the idealized stratiform boundary layer cloud model. But they leave it unclear what the difference is to the previous adiabatic assumption. Do they include a profile of sub-adiabaticity as Boers et al. (2006)? Is CDNC in this model not vertically uniform and/or the liquid water content not linearly increasing with height? What is the assumption on sub-adiabaticity?**

We have clarified this more now in Section 2.2. We use a linear profile, with condensation rate calculated from cloud top temperature at 80 % of its adiabatic value. Please note, that we introduce the terminology ISBLC in order to avoid the confusion we were facing over the last ten years when using the term 'adiabatic cloud model'. We are not defining a new cloud model. Rather, we are trying to avoid ambiguities that are embedded in the word 'adiabatic'. Our experience was that half of the community accepted the term and the other half very strongly rejected it because the model is actually sub-adiabatic. This led to long and in our view possibly pointless discussions about terminology, which we hope to avoid.

**2. The uncertainty analysis is highly superficial. Fundamentally, the authors just write that in a single case (VOCALS, 20 flights) the "error" as propagated from the MODIS- retrieved reff and tau uncertainty assessed by Platnick et al. (2015) is similar in mag- nitude as the spatial variability at the scale considered for these cases (up to 51x51 km2 ). As such, it is highly astonishing that the authors take the spatial variability at face value as the uncertainty for any other cloud regime as well. The result of course is foreseeable: the variability is small for stratiform, and large for broken clouds. Although it is not unlikely that the actual error behaves like this, it cannot be concluded from the analysis by Bennartz and Rausch. I suggest the authors either perform a rigorous uncertainty analysis, or else abstain from calling the variability "uncertainty", but actually call it, e.g., "sub-scale variability".**

We agree with the reviewer's comment about the lack of detail on uncertainty analysis in the first version of the manuscript. We have added more detail and an additional figure on uncertainty analysis now.

**Title: clarify this is only over ice-free oceans (maybe in the abstract is also sufficient)**

We added this to the abstract.

**p2 l 5: Studies preceding the Bennartz (2007) satellite climatology should be acknowl- edged, such as Han et al. Geophys. Res. Lett. 1998 (AVHRR) and Quaas et al. Atmos. Chem. Phys. 2006 (MODIS).**

Fixed.

**P3 l14: what is "maximum" adiabatic value here?**

The maximum condensation rate of an airparcel undergoing saturated adiabatic ascent, which is a weak function of temperature and pressure. We have clarified this in the text.

**P3 l16: probably something like "...a growing body of work has been devoted to under- standing..."?**

Fixed

**P3 l20: what is the distinction of CDNC at cloud base and the one "observed"? This should be clarified, since for the assumption of vertically constant CDNC, there seems to be no difference.**

We believe this difference adequately captured by the following sentence, highlighted in italcs in the following excerpt (Page 3 Line 18-21, original manuscript): "Furthermore, the cloud microphysical interpretation of retrieved CDNC is also not straight forward, as ultimately one would be interested in the number of cloud droplets activated at cloud base and not the number of cloud droplets observed. *Entrainment mixing processes, precipitation formation, and additional activation above cloud base can lead to differences between these two properties*."

**P4 l6: should read km$^2$**

Fixed

**p4 l8: specify what type of observations**

Fixed.

**p4 l18: \citep[e.g.,]||]{brenguier} and later citet{bennartz}**

Fixed

**p5 l3: it would be good to give the reader some insight about the aircraft-observed range in k. E.g. Freud and Rosenfeld (J. Geophys. Res. 2012) find k = 0.93, while values from Martin et al. 1994 and Pawlowska and Brenguier (Tellus 2003) go down to 0.67. This introduces an uncertainty of about 20%.**

We have mentioned this uncertainty now (and 20% is also what we have been using in our error propagation).

**p5 l16: "for the 13 years"**

FIXED

**p6 l1: better clarify: "liquid water cloud"**

FIXED

**p6 l28: some more detail is required how this series of reflectances was constructed. Which thermodynamic profiles, cloud-base heights and cloud-base updraft speeds / CCN concentrations were sampled?**

We have added detail to this discussion. However, since we did not start with CCN concentration, updraft speeds and CCN concentration were not considered.

**P7 l21: "horizontal and vertical" ?**

FIXED

**P7 l26: black cross? And CDNC = 500 cm-3 / H = 175 m?**

FIXED

**P8 l5: It would be useful if the authors discussed how often the MODIS algorithm fails to diagnose partly cloudy pixels, i.e. how important this problem remains after condition (4) on p6 is enforced.**

This issue is addressed in Section 4 in much detail both in terms of its impact on mean CDNC as well as in term of its impact on data density.

**P8 l12: correct color and numbers p8 l13: "an optical depth"**

FIXED

**p8 l15: the assumed single scattering albedo should be reported**

The single scattering albedo at 0.55 micron (SSA) reported by Haywood is 08.9-0.91. The cont. poll. SSA is 0.892. These values of course depend on wavelengths. The full information can be looked up in the referenced publications, which we believe is sufficient for the study presented here.

**p8 l16: why "not shown"? It is represented in Fig. 1 as well, as far as I understand.**

Good point. FIXED.

**P9 l6: "angle increases" or "angles increase"**

FIXED

**p9 l16: the screening criteria should be explained (reference to Tab. 3)**

FIXED

**P9 l20: "from"**

FIXED

**p9 l22: "in a manner that"**

FIXED

**p10 l2: "biases" - it is not a priori clear that these are biases, or do I miss a point?**

We have formulated this more carefully now.

**P10 l15: citet**

FIXED

**p11 l2: The authors should report how many datapoints are sorted out by the stratification criterion.**

We added this information.

**P11 l7: From Fig. 4, it seems the problem of reduction in amount of data is mainly due to the sunglint angle > 35 criterion. However, compared to the scattering angle criterion, this one seems to be of minor importance. I suggest to split the two issues and investigate them separately.**

While the sunglint angle criterion does remove a large amount of data, we could not detect any significant sensitivity of the results toward including or excluding this data. We have tried to make this

argument clearer now in the paper. It really is the scattering angle that determines systematic artifacts we are seeing. This would also make sense from the standpoint of the retrievals. Both, solar zenith angle and sunglint angle are partly correlated with scattering angle, so they do show some of the same effects because of this correlation.

**P12 l20: This is a very brief uncertainty quantification discussion. The notion that spatial variability could fully represent the error seems implausible.**

We agree the uncertainty quantification was too short. See our general comments further up.

**P13 l22: the trend should have a per time unit.**

FIXED

**P13 l28: aerosol indirect effect? Or rather "radiative forcing due to aerosol-cloud inter- actions"?**

We were thinking of the first indirect aerosol effect here

**P14 l25: traceability**

FIXED

**p20, Fig. 2: It would be desirable to add smaller CDNC to the plot, since CDNC < 20 cm-3 are frequently retrieved.**

We do not believe this will add value to the analysis presented here, since we are exemplarily showing results for one observation geometry and under highly idealized conditions. If a similar method were used in a true retrieval, we agree the lower limit should be changed.

**P21, Fig. 3: The caption should explain the label "fraction of open water": is this 1 – cloud fraction? Also the scale is unclear, is this at the 1x1 km2 scale?**

We have added a sentence to clarify this.

**P22, Fig. 4: Are the curves averaged over the year and globe for given scatter- ing/sunglint angle?**

Correct. We have revised the caption slightly in order to make this clearer. See also additional discussion further up.

**P23 Fig. 5: title top panel "Difference"**

FIXED

**p26 l8: this is the central result of the climatology, so it deserves more attention. I suggest to move the boxes from the top panel to the (less important) bottom panel.**

FIXED

**Rather than reporting the number of missing months, I suggest to report the fraction of missing days in each 1 x1 grid box. For the climatology (top panel) a color should be chosen that better allows to distinguish CDNC at lower concentrations, possibly a non-linear color scale would be very helpful in this regard.**

We re-plotted CDNC on a log-scale as suggested. We kept the missing months for the third panel as we feel this information is more useful to the user of the climatology, since it more directly relates to the actual content of the climatology.

p29 references: journal names should  be abbreviated. Some  titles are in upper  cases.

FIXED

Response to Anonymous reviewer 2

We thank both reviewers for their in-depth reviews and their thoughtful comments. In response to the reviewers' main criticism we have added another section and figure dealing with uncertainty estimates and outlining our argument for why we think the standard deviation is the better uncertainty estimate. Below please find our point-by point response to the reviewer's concerns.

This paper describes a climatology of monthly mean cloud droplet concentrations (CDNCs) using the MODIS-Aqua satellite instrument. Some of the potential retrieval issues that can affect CDNC retrievals are discussed along with some examples of maps and seasonal cycles from the final climatology. This data set is likely to be a very useful commodity to many researchers and I commend the authors for making it available online. Thus I recommend the publication of the paper once certain issues have been addressed and additional discussion points added. The major issues are summarized next with more specific comments and typos below that.

**Major issues**

One key point on which I disagree with the authors is their restriction to situations where re,3.7 > re,2.1 > re,1.6. Painemal and Zuidema (2011) showed that this criteria was often violated for MODIS retrievals of stratocumulus (generally re1.6, re2.1 and re3.7 were very close) and yet the aircraft observed LWC was found to increase with height. This is likely due to other errors in the retrieval of reff that affect the different MODIS channels in different ways. E.g. sub-pixel heterogeneity can increase re2.1 more than re3.7 even in relatively homogeneous stratocumulus (Zhang, 2012), and Grosvenor (2014) showed that resolved heterogeneity is likely to affect the re3.7 more than re2.1. Also, King (2012) show that the information content from the different MODIS channels is not enough to be able to retrieve vertical height variation in reff and so it seems to me that this restriction might be causing some spurious filtering of the data. It is also not clear from Fig. 3 that re3.7 > re2.1 would select the cloud covered part of the curve as stated since re3.7 and re2.1 are very close, as was also seen in the Painemal and Zuidema (2012) aircraft observations. Differences between the stratified and non-stratified CDNC values are reported in Section 4.2.1, but can we be sure which is more realistic? I think that there should be more discussion about and acknowledgement of these issues.

Thanks for pointing us to this. We agree and we have addressed this issue in the paper as well as in the conclusions now.

Consideration of scattering angle effects and the effect of sunglint is made. However, the applied filtering does not take into account the likelihood of retrieval biases at high solar zenith angles. A little more detail should be given on this issue along with consideration on how this bias will affect your data set. For example, this could potentially cause large-scale biases in the winter months at high latitudes.

It can't be said for sure whether some of the scattering angle and sunglint effects in Section 4.1 are due to variations in region – certain regions or seasons may be sampled more often at certain scattering angles and with certain sunglint angle. And there can be strong natural regional changes in reff, tau, CDNC, etc. Is there a way to rule this out? E.g. can you look at the differences from within the same swath relative to the swath mean and then average these, so that the locations of samples are roughly the same? This is possible for sensor angle (as in Maddux, 2010) and sunglint angle, but I'm not sure

about scattering angle. There is also no direct examination of the effect of at sensor angle. This would be useful or at least you could directly refer to this from the results of Maddux (2010)? Also, the swath edges may also sample a different range of scattering angles than elsewhere in the swath, which may also cause artifacts. Have you looked into this?

Yes. We have looked into the issue regionally and the scattering angle dependency comes out very clearly regardless of where you look. We believe this is an inherent issue related to the actual retrieval code rather than something related to natural variability or unfavorable observer or solar zenith angles. We have tried to clarify this in the paper but we do not think that it is necessary to add further plots to it as the effect is clear even on a global scale.

In this regard we also feel that an expansion of the section to cover every possible situation of observer and solar zenith angles will not add to what we already know from the work of e.g. Maddux and Grosvenor. We believe the issue we are concerned with much more an algorithmic that could be remedies also in a plane-parallel world if the retrieval algorithm accounted correctly for single scattering. Further, if the angular dependencies in itself are of interest, it would make more sense to study these effects in the context of the actually retrieved effective radius and optical depth, rather than in the context of CDNC.

RE "uncertainties" – using the variability of the daily CDNC values means that a lot of what is termed "uncertainty" will actually be natural variation in CDNC, which, even with perfect retrievals, would not be expected to be constant throughout a cloud field, nor over the course of a month, nor over the climatology. I understand that given the fact that the uncertainties are not well characterized it is hard to separate the real variability from the noise introduced by retrieval errors. However, perhaps the quoted term should just be renamed "standard deviation" with it being made clear that some of this will be actual variability and some uncertainty. Also, since you can calculate the propagated CDNC uncertainty from the pixel level uncertainties in tau and reff and I think it would be useful if these were provided to get a sense of how much they contribute. However, it should also be made clear that these would not account for the entirety of the uncertainty since there will be a lot of error introduced by the forward model plane parallel independent pixel approximation that will not be captured.

We agree with the reviewers that this was discussed too superficially in the original version of the paper. We have added a figure and a section discussing uncertainty in more detail. We do believe that the term 'uncertainty' is the correct term though, as we are concerned with the monthly mean CDNC values.

Fig. 5 - it looks like there is very little data in box X12 – the numbers of samples going into each box should be quoted for this figure. There is also overlap between box R03 and X12, yet box X12 shows quite different results, perhaps indicating that the statistics for this box are too poor to be robust?

Your notion about X12 and R03 is correct but it really is caused by differences in the annual cycle. If you look at Fig 10. Bottom panel, you can see how different the annual cycles are in this area. Since R03 is larger and further south, it sees a much different annual cycle than X12. In retrospect, R03 is probably an ill-defined and too wide grid box.

We are not too concerned with the data density. Recall that only grid-boxes with at least ten days per month with at least 10 observations each make it into the climatology. Those were then averaged to get the average values for each of the larger boxes like X12. Thus each box has virtually hundreds of observations in it. Also, for each of the boxes, the annual cycle is relatively smooth. If we had a random noise problem, this would also show in very noisy annual cycles.

**Specific comments**

**Abstract** – **"Resulting CDNC uncertainties for the climatology are in the order of 30% in the stratocumulus regions and 60% to 80% elsewhere"** – as discussed in the other comments is it appropriate to call this "uncertainty". A more careful description is needed here.

We hope to have addressed the issue by providing more in-depth discussing. We do believe the term uncertainty to be appropriate though.

**p.2, L27** – **"1. The cloud is assumed to be horizontally homogeneous."** – This seems a little vague. It would be good to mention over what scales the cloud needs to be horizontally homogeneous and why. E.g. 1km, 1x1 degree or both? Or larger even? Perhaps it would be worth mentioning the application of the independent pixel approximation for reff and optical depth retrievals such that inhomogeneity over scales larger than the 1km pixel size is likely to violate this assumption with the scale being set by the degree to which net horizontal photon transport occurs (the scale of any shadows/bright spots, etc., or rather deviations from the plane-parallel reflectances, as defined in Marshak, 2006). This could be separated from the requirement of homogeneity within the 1km pixel to satisfy the plane-parallel retrieval assumption. In terms of the CDNC derivation itself only the sub-pixel homogeneity is explicitly required once we have the correct 1km reff and optical depth since 1km resolution CDNC values can be calculated.

Thanks for pointing this out. This refers to clouds at Level 2 at a scale of about 1x1 km. We have clarified this now.

**p.3, L16** – **"The true three-dimensional variability of clouds poses significant challenges to any remote sensing algorithm and a growing body work has been devoted toward understand the impact of this variability on remote sensing estimates of CDNC."** – these works should be cited here, or else you should refer to them in another section in the paper.

FIXED

**p.3, L19 – "as ultimately one would be interested in the number of cloud droplets activated at cloud base and not the number of cloud droplets observed"** – I think this would depend on the application. Some studies may be interested in how the cloud top CDNC might change due to lateral mixing, evaporation, etc., or removal of CDNC by precipitation and not necessarily just the cloud base CDNC. I can see that the cloud base CDNC would be of interest for comparing to model processes, but I think that the statement here generalizes too much.

Agreed. FIXED

**p. 4, Eqns. 2 & 4** – You should describe how cw (condensation rate) was calculated – did you use the MODIS cloud top temperature? What pressure did you use? Did you use the full adiabatic value, or 80% of this as mentioned on p.3. An issue with the calculation used in Bennartz (2007) is mentioned in the acknowledgments - it would be good to add more detail in the main part of the paper about this issue along with a reference for the correct formula – e.g. Ahmad (2013). A discussion of the compensation of errors in k, fraction of adiabatic condensation rate, reff errors found by Painemal and Zuidema (2012) should also be included.

We have added some discussion and a reference.

**p.6 – "3. The cloud mask had to indicate the observation to be cloudy but not over ice or land."** – does this include the filtering of sea-ice covered regions? How robust is the detection of sea-ice? This is important since this is likely to lead to a poor retrieval.

Agreed. The ice mask would be important near the ice edge, but we do not know the answer to this question. We have clarified in the text that we are only concerned with ice-free ocean.

**p.6 – "6. Observations were only considered, if the three MODIS-retrieved effective radii stacked up as re,3.7 > re,2.1 > re,1.6 ,as observations violating this criterion will also violate the key assumption of a vertically increasing LWC in the ISBLC."** – this is a key point on which I disagree with the authors – Painemal and Zuidema (2011) showed that this criteria was often violated for MODIS retrievals of stratocumulus (generally re1.6, re2.1 and re3.7 were very close) and yet the aircraft observed LWC was found to increase with height. This is likely due to other errors in the retrieval of reff that affect the different MODIS channels in different ways. E.g. sub-pixel heterogeneity can increase re2.1 more than re3.7 even in relatively homogeneous stratocumulus (Zhang, 2012), and Grosvenor (2014) showed that resolved heterogeneity is likely to affect the re3.7 more than re2.1. Also, King (2012) show that the information content from the different MODIS channels is not enough to be able to retrieve vertical height variation in reff and so it seems to me that this restriction might be causing some spurious filtering of the data.

Thanks for pointing us to this. We agree and we have addressed this issue in the conclusions now.

**p.6, L24 – "a solar zenith angle of 56 degrees. The resulting scattering angle of 124 degrees"** – it would be good to define what you mean by the scattering angle. I.e. that it is the angle between the sun and the satellite within the plane of the sun and the satellite and that 180 degrees is backscatter.

We added this information.

**p.7, L23 – "Typically, most of these issues affect the effective radius at 1.6 µm and 2.1 µm more strongly than the effective radius at 3.7 µm (Zhang et al., 2012b)."** – this was true for the sub-pixel optical depth heterogeneity effect, but are there other examples? Grosvenor (2014) showed that the 3.7um channel is likely to be more strongly affected by resolved 3D radiative effects, so this is not always the case. Although the latter effect is more likely at higher solar zenith angles.

We added this information and reference.

**p.7, L33** – "observations (i.e. if cloud thickness and liquid water path varies within the field-of-view)."
– Strictly, Zhang (2012) talk about the effect of having sub-pixel heterogeneity in optical depth (which cloud arise through combinations of LWC, CDNC, etc.) changes, rather than cloud thickness and LWP.

We were not sure how this statement is different from what we wrote.

**p. 8, L 4** – "so that in the example shown in Figure 3 (right panels) the largest part of the retrievals would be (correctly) rejected." – it is not clear from Fig. 3 that re3.7 > re2.1 would select the cloud covered part of the curve since re3.7 and re2.1 are very close, as was also seen in the Painemal and Zuidema (2012) aircraft observations. Why not just filter based on the cloud fraction?

Sub-scale cloud fraction is not available for the MODIS observations.  This is a theoretical experiment how the (in reality unknown) sub-scale cloud fraction affects the retrievals.

**p.9, L4** – "Grosvenor and Wood (2014) address issues the dependency of cloud microphysical retrievals on solar zenith angle." – However, it seems that the filtering applied does not take into account the likelihood of retrieval biases at high solar zenith angles and this issue is not brought up again. A little more detail should be given on this issue along with consideration on how this bias will affect your data set.

Agreed. We added this disclaimer in the conclusions.

**p.9, L14 & Fig. 4** – sunglint angle is not defined. I'm guessing that you mean the absolute difference between the sensor zenith angle of a given pixel and the sensor zenith angle of the middle of the sunglint band within the swath, but it is not obvious.

We added the definition of sunglint angle.

**Section 4.1** – It can't be said for sure whether some of the effects seen are not due to variations in region – certain regions or seasons may be sampled more often at certain scattering angles and with certain sunglint angle. And there can be strong natural regional changes in reff, tau, CDNC, etc. Is there a way to rule this out? E.g. can you look at the differences from within the same swath relative to the swath mean and then average these, so that the locations of samples are roughly the same? This is possible for sensor angle (as in Maddux, 2010) and sunglint angle, but I'm not sure about scattering angle.

Regarding angular dependencies: Please see our answer at the very top.

**p.10, L10** – presumably high sunglint angles (if my definition of this is correct) will be associated with high sensor angles since they will have to be near the edges of swaths. So, this could be contribute to the artifact. Why not also look at sensor angle directly? Or at least directly refer to it from the results of Maddux (2010)? However, the above interpretation seem inconsistent with Maddux (2010). They showed that the swath edge liquid optical depth values were lower and the re higher than those near nadir. Are the results consistent with Maddux (2010) as stated at line 15? Also, the swath edges may also sample a different range of scattering angles than elsewhere in the swath, which may also cause artifacts. Have you looked into this?
Regarding angular dependencies: Please see our answer at the very top.

**Section 4.2.1** – differences between the stratified and non-stratified CDNC values are reported, but can we be sure which is more realistic? Also, it looks like there is very little data in box X12 – the numbers of samples going into each box should be quoted for Fig. 5. There is also overlap  between box R03 and

See our answer to identical point above under Major comments.

**Fig. 6 –** the title uses the term "unflagged", but "stratified" would be more consistent with the rest of the paper. Also, the frequent positive values seem inconsistent with that expected from Fig. 4 where the main effect appears to be a decrease in CDNC at high scattering angles. Or is the increase at low sunglint angles dominating for these regions? Why not also exclude the very high sunglint angles for the flagged pixels?

We have corrected the terminology. It is very hard to find the expected value for difference from Fig 4., because of the large differences in frequency of occurrence of different angular combinations. High sunglint angles are not excluded because they do not pose a retrieval challenge (virtually no direct backscatter from the ocean surface).

**p.12, L 5 – "PZ11. Various effects, including cloud top entrainment or the representativeness of the chosen condensation rate, can potentially cause the somewhat larger bias for the stratified cases."** – It's hard to see how this might be the case? Can you explain? Could it also be that the stratified cases might be further from the aircraft observation?

You are right. That was a weak explanation. We tentatively agree with the point you made in the very beginning and have revised the explanation accordingly.

**p.12, L10 – "However, for both 51x51 and 21x21 neighbourhoods, the mean uncertainty (size of error bars in Figure 7 and last two columns in Table 1) is reduced for stratified cases over the un-stratified cases. … Effective radius stacking is likely decreasing the selection of pixels of inhomogeneous clouds or those subject to sub-pixel effects, which, as discussed in Section 3, can result in retrievals with a wide spread of unphysical effective radii."** - Could this also be due to there being fewer samples in each region due to being more selective (since this would reduce the std. deviation)? Or were all boxes chosen to have all possible pixels present?

It might just be because of the suppression of noise due to the selection criterion.

**p.13, L3 –** RE "uncertainties" – using the variability of the daily CDNC values means that a lot of what is termed "uncertainty" will actually be natural variation in CDNC, which, even with perfect retrievals, would not be expected to be constant throughout a cloud field, nor over the course of a month, nor over the climatology. I understand that given the fact that the uncertainties are not well characterized it is hard to separate the real variability from the noise introduced by retrieval errors. However, perhaps the quoted term should just be renamed "standard deviation" with it being made clear that some of this will be actual variability and some uncertainty. Also, since you can calculate the propagated CDNC uncertainty from the pixel level uncertainties in tau and reff and I think it would be useful if these were provided to get a sense of how much they contribute. However, it should also be made clear that these would not account for the entirety of the uncertainty since there will be a lot of error introduced by the forward model plane parallel independent pixel approximation that will not be captured.

We have added significant discussion and clarification on the issue of uncertainty. We have also, as suggested, provided in Figure 8 quantitative analysis on the error-propagation results.

**p. 13 & Fig. 9 – "magnitude of the annual cycle"** – can you state somewhere what you actually mean by this? Is it the peak-to-peak amplitude of the annual cycle (i.e. max monthly mean minus min monthly mean), or is it calculated based on the cosine fits? "Amplitude" would be a better word than "magnitude" since it relates more directly to the definition.

Good point. We have revised this to amplitude and specified we mean the amplitude of the cosine fit.

**Fig. 10** – the bottom right panel needs the y-axis range changing to show the negative values.
FIXED

**p. 14, L31** – "We further show that neglecting this screening does not only lead to moderate biases in the annually averaged CDNC" – Are these biases or just differences? How do we know which one is correct due to the issues raised earlier in this review?
Agreed. We changed this to read differences.

**p.15, L32 – "We found some remaining retrieval artefacts in the MODIS-retrieved effective radius and optical depth that propagate through into artefacts of the CDNC climatology as well. "** – it would be useful to mention the issues that you are talking about here.

Done

**p.15, L6 – "However, feel that it might be beneficial in future work to re-create retrievals working directly on the Level-1 reflectances that address some of the issues we have identified."** – although, perhaps it should be mentioned that it may not be possible to resolve many of these issues even if re-processing Level-1 data – e.g. it would be hard to correct biases for 3D radiative effects, sunglint effects, etc.

Done.

**Tables –** it would make sense to alter the order of these to align with the order of mention in the paper – i.e. first Table 3, then Table 2 and then Table 1.

FIXED

**Typos**

p.9, L4 – "Grosvenor and Wood (2014) address issues the dependency of" – remove "issues".

FIXED

p.9, L20 – "Fro Figure 4"

FIXED

p.9, L22 – "vary in a manner the roughly resembles"

FIXED

p.10, L3 – "Since Aqua's local equator crossing time around 14:30,"

FIXED

p.10, L13 - "with an strong increase retrieved optical depth" -> "with a strong increase in retrieved optical depth"

FIXED

p. 10, L19 – "4.2. Impact Stratification and retrieval artefacts on climatology" -> "Impact of Stratifcation…"

FIXED

p.10, L24 - "boxes are shown were all 12 months"

FIXED

p.12, L2 – "in the order of" -> "on the order of"

FIXED

p.12, L4 – "For stratified results, magnitude of the RMSE and bias increase" -> "For the stratified results, the magnitude …"

FIXED

p.14, L25 – "a better traceable of results" –> "a better traceability of results"

FIXED

p.15, L6 – "However, feel that"
FIXED

p.20, L20 – "cover have discussed" -> "have been discussed"
FIXED

**References (only ones not cited in the paper under review are listed)**

Ahmad, I., Mielonen, T., Portin, H. J., Arola, A., Grosvenor, D. P., Mikkonen, S., Leskinen, A., Komppula, M., Lehtinen, K. E. J., Laaksonen, A., and Romakkaniemi, S.: Long term measurements of cloud droplet concentrations and aerosol-cloud interactions in boreal boundary layer clouds, Tellus B, 65, 20138, doi:10.3402/tellusb.v65i0.20138, 2013.

Marshak, A., Platnick, S., Varnai, T.,Wen, G., and Cahalan, R.: Impact of three-dimensional radiative effects on satellite retrievals of cloud droplet sizes, J. Geophys. Res.-Atmos., 111, D09207, doi:10.1029/2005JD006686, 2006.

King, N. J. and Vaughan, G.: Using passive remote sensing to retrieve the vertical variation of cloud droplet size in marine stratocumulus: an assessment of information content and the potential for improved retrievals from hyperspectral measurements, J. Geophys. Res.-Atmos., 117, D15206, doi:10.1029/2012JD017896, 2012.

**Global and regional estimates of warm cloud droplet number concentration based on 13 years of AQUA-MODIS observations**

Ralf Bennartz[1,2], John Rausch[2]

5  [1] Earth and Environmental Science Department, Vanderbilt University, Nashville, TN, 37240, USA
[2] Space Science and Engineering Center, University of Wisconsin – Madison, WI, 53706, USA

*Correspondence to*: Ralf Bennartz (ralf.bennartz@vanderbilt.edu)

**Abstract.** We present and evaluate a climatology of cloud droplet number concentration (CDNC) based on 13 years of Aqua-MODIS observations. The climatology provides monthly mean 1x1 degree CDNC values plus associated uncertainties
10  over the global ice-free oceans. All values are in-cloud values, i.e the reported CDNC value will be valid for the cloudy part of the grid-box. Here, we provide an overview on how the climatology was generated and assess and quantify potential systematic error sources including effects of broken clouds, and remaining artefacts caused by the retrieval process or related to observation geometry. Retrievals and evaluations were performed at the scale of initial MODIS observations (in contrast to some earlier climatologies, which were created based on already gridded data). This allowed us to implement additional
15  screening criteria, so that observations inconsistent with key assumptions made in the CDNC retrieval could be rejected. Application of these additional screening criteria led to significant changes in the annual cycle of CDNC both in terms of its phase and magnitude. After an optimal screening was established a final CDNC climatology was generated. Resulting CDNC uncertainties are reported as monthly-mean standard deviations of CDNC over each 1x1 degree grid box. These uncertainties 
[revised manuscript text omitted]
 recent studies have made progress toward understanding and possibly correcting the impact of this variability on cloud remote sensing (Zhang et al., (2012), Zhang et al. (2016) and references therein). Furthermore, the cloud microphysical interpretation of retrieved CDNC is also not straight forward, as ultimately one would be interested in the number of cloud droplets activated at cloud base and not the number of cloud droplets observed. Entrainment mixing processes, precipitation formation, and additional activation above cloud base can lead to differences between these two properties. We will elaborate more on issues related to three-dimensional cloud structure as well as cloud microphysical assumptions in Section 3.

The remainder of this paper is structured as follows. In Section 2 we briefly describe the datasets as well as the principal methods used here to derive CDNC from satellite observations. Section 3 summarizes issues related to the use of the ISBLC as well as other assumptions made in the retrieval process, thereby providing interpretational context for the use of satellite-derived CDNC retrievals as well as guidance on the expected magnitude and relative importance of uncertainties introduced by the various assumptions. In Section 4 we address actual uncertainties and possible biases in CDNC retrievals. Potential biases are for example caused by remaining artefacts in the underlying MODIS retrievals caused by unresolved dependencies on observation geometry or underlying assumptions on the width of the droplet spectrum. In addition, in Section 4 we also validate our CDNC retrievals against in-situ observations of CDNC taken during the VOCALS-Rex campaign and summarized by Painemal and Zuidema (2011). In Section 5 we evaluate the 13-year climatology of MODIS observations. In

[revised manuscript text omitted]

5   cases, as shown below.

**4.4 Uncertainty estimates**

As out lined in Section 2.3, two main pathways exist for the definition of uncertainties associated with the climatology. Firstly, uncertainties in retrieved effective radius and optical depth reported in the MODIS Collection 6 Level 2 product can be propagated forward to yield Level 2 uncertainties in CDNC (see Bennartz, 2007 for details). Secondly, the standard

10   deviation of all Level 2 observations within a given 1x1 degree box can serve as a measure for the product's uncertainty. The latter would include the synoptic variability of CDNC as well as all actual retrieval uncertainties and should in any case be larger than the uncertainty derived from error propagation. Here, we compare these two measures. Recall further from Section 2.3 that the monthly uncertainties for each grid-box were calculated as the square-root of the mean of the daily uncertainties over the course of each month, thereby following the approach used in NASA's Level-3 gridded MODIS cloud

15   product (Hubanks et al., 2016). Figure 8 compares the two different uncertainty estimates in terms of the relative uncertainty averaged over the entire year 2008. Except for some stratocumulus regions the uncertainty estimate based on the standard deviation of the individual Level 2 observations is significantly larger than the uncertainty based on error propagation. This would be consistent with monthly uncertainties driven by synoptic variability.  However, for some of the stratocumulus areas, outlined with black isolines in the bottom panel of Figure 8, the uncertainty by standard deviation is smaller  (by up to

20   50%) than the uncertainty by forward propagation of retrieval errors.  Hypothetically, even if the actual CDNC in those areas were perfectly constant spatially and temporally, the uncertainty by standard deviation would be identical to the uncertainty by forward-propagated retrieval error. The only scenario under which this situation could be reversed would be, if estimated uncertainties feeding into the theoretical forward propagation are too large. For example, if the reported uncertainty of the effective radius in the MODIS Level 2 product were too large, this would lead to too large uncertainties in the propagated

25   CDNC uncertainty. Thus, from the bottom panel of Figure 8, it would appear that the forward propagated uncertainties for individual CDNC retrievals are slightly too large. That is, we put slightly less trust into the Level 2 observations than appears to be warranted by this comparison.

For the practical matter of characterizing uncertainty associated with CDNC retrievals for the gridded product, the reported uncertainty estimates should include the impact of day-to-day variability in order to allow for a realistic estimate of the true

30   variability of CDNC.  In the final CDNC climatology we therefore report the standard deviation of the retrieved CDNC values within each 1x1 degree grid box as an uncertainty estimate.

**5. Climatological results**

**5.1. Global overview**

Figure 9 and Figure 10 show a global overview of the CDNC climatology. The upper two panels of Figure 9 show mean CDNC and relative uncertainty for the entire time period. The lowermost panel of Figure 9 shows the fraction of months with missing data. One can see that mean CDNC is typically high near the coasts, in particular downwind of the major continents, and lower over the remote oceans. In particular in the tropics certain areas exhibit very low data coverage. Areas in the northern Indian Ocean and the western Pacific exhibit no valid data at all. These areas are typically associated with convective regions where the ISBLC assumption frequently breaks down because either isolated cumuli or deep convection are observed. Near the fringe of these regions also the largest relative uncertainties are observed. Over the mid-latitude storm tracks as well as over the traditional stratocumulus areas west of the major continents, data coverage is higher and various regions show full coverage for all months. Relative uncertainty is in the order of 60% to 80% in the storm tracks and increases polewards. In the stratocumulus regions the relative uncertainty is about 30%. Recall that uncertainties are calculated as the square-root of the mean of the daily 1x1 degree variances over the course of each month, thereby assuming uncertainties between days to be uncorrelated (see Section 2.3). The reported uncertainties are however very similar to the a-posteriori uncertainties derived from error propagation (see Section 4.3 and 4.4).

Figure 10 shows information on the annual cycle of CDNC. The data density in Figure 10 is lower than in Figure 9 because we only included grid boxes where a full annual cycle was available. For some areas, in particular in the tropics, data was only available for certain months and these grid boxes were excluded from the annual cycle analysis. The upper panel of Figure 10 shows the amplitude of the annual cycle both color-coded and as isolines. Amplitude here is defined as the amplitude of a cosine-fit of the annual cycle of CDNC. 
[revised manuscript text omitted]
. Potentially adverse effects of this screening could occur in situations where the three effective radii are very close to each other, as might be the case for thin stratocumulus clouds. In such cases random noise in the observations might remove valid observations and potentially bias CDNC slightly low as preferably higher effective radii at 3.7 μm would be selected. This hypothesis would be consisntent with our finding comparing to the PZ11 cases. However, we believe this issue secondary compared to the large number of situations where the radiation field is

20  affected by either broken clouds or precipitation and the three effective radii clearly indicate that key assumptions in the CDNC retrieval are violated. Further retrieval issue might arise at high zenith angles and/or near the ice edge. At high zenith angles, earlier work by Grosvenor and Wood (2014) showed the effective radius at 3.7 μm to be more strongly biased, potentially leading to retrieval issues. Further, cloud mask classification errors near the ice edge might lead to increased noise and our artefacts in those regions.

25  We found some remaining retrieval artefacts in the MODIS-retrieved effective radius and optical depth that propagate through into artefacts of the CDNC climatology as well. These artefacts manifest itself in a strong dependency of retrieved effective radius on scattering angle and could potentially be related to the treatment of first order scattering in the MODIS Level 2 retrievals of effective radius and optical depth. Our assessment of those effects on the climatology shows however that effects clearly visible in the Level-2 data are averaged out to some degree in the aggregated climatology. We therefore

30  did not in the final climatology include any additional screening for those effects. However, we feel that it might be beneficial in future work to re-create retrievals working directly on the Level-1 reflectances that address some of the issues we have identified. Such retrievals could be specific to stratiform boundary layer clouds and might include more appropriate

[revised manuscript text omitted]

---

## Referee Report (RR1)

**Review of v2 of Bennartz and Rausch – "Global and regional estimates of warm cloud droplet number concentration based on 13 years of AQUA-MODIS observations"**

We thank the authors for addressing most of the comments. I feel that a few of the issues need a bit more attention, though :-

**Uncertainty analysis**

The new section does somewhat clear up the matter of the uncertainty vs variability, particularly for the stratocumulus regions where the overall variability (i.e. actual variability + instrument uncertainty) is smaller than the instrument uncertainty alone. It is now helpful that the meaning of the uncertainty is made clear in the paper.

Although, this raises the issue of systematic errors/offsets vs random uncertainties. The use of the instrument uncertainties from MODIS (which are just the radiance uncertainties propagated through to reff and tau) neglects uncertainties in the forward model relating to heterogenetiy, etc. This might produce a fairly constant offset error in Nd, which would therefore not show up as variability in the standard deviation. In the current paper the propagation of errors in reff, etc. through to Nd that was done in Bennartz (2007) is mentioned – such an analysis might do a better job of estimating such offset errors to give a better estimate of how far off the quoted Nd values are from reality. However, they don't seem to be used in the current work. I think that it would be good to mention the possibility of such "offset errors" and to quote the uncertainty range calculated in Bennartz (2007).

**Solar Zenith Angle dependence**

p.7, L31 – "Conversely, at high solar zenith angle also the the effective radius at 3.7  $\mu$ m might also be biased leading to possible increases in CDNC by 40% to 70% at solar zenith angles higher than about 70 degrees (Grosvenor and Wood, 2014)."

This doesn't quite address what I was trying to convey with my review comment. Firstly, Grosvenor and Wood (2014) showed that optical depth biases were mainly responsible for the 40-70% increase in CDNC. I think that a sentence like this should be moved to p.9, L15 where you talk about the view geometry biases. There, instead of "Grosvenor and Wood (2014) address the dependency of cloud microphysical retrievals on solar zenith angle.", you could perhaps write :-

"Grosvenor and Wood (2014) address the dependency of cloud microphysical retrievals on solar zenith angle demonstrating a possible increases in CDNC by 40% to 70% at solar zenith angles higher than about 70 degrees."

Then on p.7 something like this would be more akin to what I meant :-

"However, at high solar zenith angles Grosvenor and Wood (2014) demonstrated that resolved (as opposed to sub-pixel) 3D radiative effects are likely to cause the effective radius to be biased high, with larger biases expected for the 3.7  $\mu$ m retrieval compared to the 1.6  $\mu$ m one."

**Sampling in regional boxes**

From your response:-

"We are not too concerned with the data density. Recall that only grid-boxes with at least ten days

per month with at least 10 observations each make it into the climatology. Those were then

averaged to get the average values for each of the larger boxes like X12. Thus each box has

virtually hundreds of observations in it."

However, there may be few 1x1 degree grid boxes that are included in the region X12 for a given month, and this will vary by month. It would be useful to check what these numbers look like and to quote them in the paper.

**Issues RE "Specific comments"**

*p.3, L19 – "as ultimately one would be interested in the number of cloud droplets activated at cloud base and not the number of cloud droplets observed" – I think this would depend on the application. Some studies may be interested in how the cloud top CDNC might change due to lateral mixing, evaporation, etc., or removal of CDNC by precipitation and not necessarily just the cloud base CDNC. I can see that the cloud base CDNC would be of interest for comparing to model processes, but I think that the statement here generalizes too much.*

This does not seem to have been addressed.

Typos p.5, L6 - "MODI derived".

p.6, L11 – No need for the commas here.

P13, L7 – "Out lined"

Fig. 8 seems to have no caption.

p.16, L18 - "consisntent"

p.16, L24 – "our artefacts"

p.16, L26 - "itself" should be "themselves".

P17, L4 - "by directly" should be "directly by"

p.16, L21 :- "Further retrieval issue might arise at **high zenith** angles and/or near the ice edge. At **high zenith** angles, earlier work by Grosvenor and Wood (2014) showed the effective radius at 3.7 μm to be more strongly biased, potentially leading to retrieval issues."

- should include "solar" before "zenith" in both instances.

p.12, L25 – "observe" should be "observed".

---

## Author Response (AR2)

**1. Response to both reviewers**

We want to thank both reviewers for their very thorough and insightful reviews. They have clearly helped make the paper better and clarify several issues. We have adapted most of the reviewers' comments. There are a few points were we still disagree with the reviewers. Those are discussed in detail and hopefully clarified below.

**1.1. Systematic errors versus random uncertainty**

Both reviewers have raised the issue of systematic errors versus truly random uncertainties. We agree that possible systematic errors in CDNC-retrievals are at a minimum as important as an understanding of natural variability or random errors caused by e.g. sensor noise. In fact, we devote large parts of the paper to understanding systematic errors associated with the retrieval. Sections 3.3, 3.4, 4.1, and 4.2 and many references therein deal with possible systematic error sources. Clearly, the dependency of retrieved effective radius on scattering angle illustrated in Figure 4 is a source of large systematic error for individual CDNC retrievals. We then go on to addressing the impact of this systematic error on the monthly mean climatology as presented in Section 4.2.2. We find that this particular source of systematic error (scattering-angle dependency of effective radius retrieval) only has a relatively weak effect on the climatology that can also be quantified.

We might consider some other effects maybe as 'Known unknowns'. For example, sub-grid scale inhomogeneities, variations in condensation rate, or variations in the degree of sub-adiabaticity are known to be affecting the retrieval. If their effect or magnitude was fully understood, they could be quantified or corrected for.

However, the best we can do at this point, is establish a best estimate and uncertainty range and then use error propagation to estimate what the impact of those factors is. This was done in Bennartz (2007) and again in the current publication. As both reviewers point out, we have perhaps not clarified well enough what we did in the error propagation section of the current paper and have therefore revised this section and added information as requested by both reviewers.

In a broader sense, what we are lacking though is some sort of 'ground truth' or independent validation (with the exception of the few existing highly useful joint in-situ/remote sensing observations such as VOCALS). We discuss this point in the conclusion section: Firstly, there is a need for independent remote sensing observations of CDNC using methods different from the vis/nir methods used here. Secondly, there is a need for further collocated airborne in-situ and remote sensing observation that allow assessing CDNC retrievals against an independent 'standard'.

**2. Response to Reviewer 1**

**2.1. Major comments**

**2.1.1. ISBLC Terminology**

**Reviewer Comment:**

> As for the "Idealized Stratiform Boundary Layer Cloud", I believe I'd be happy if they explicitly wrote, at the instance where the authors introduce this term, that they mean exactly the same they meant before by the term "adiabatic cloud model". As such, the reader can understand that it is simply a re-branding of the same thing.

**Reply:** We believe that we already made this point clear when we introduced the term ISBLC (Line 6-13, page 3). We state that this is a question of "terminology" and that "we refrain from using the term (!) 'adiabatic cloud model'". Then, we go on to state that: "A term (!) that probably better, albeit less attractively, describes the intent of the above assumptions would be the 'Idealized Stratiform Boundary Layer Cloud' (ISBLC) model, which we will use throughout this paper." We believe this captures exactly what the reviewer is asking for.

**2.1.2. Use of the term 'uncertainty'**

**Reviewer Comment:**

> As for the "uncertainty" assessment – the authors still call the sub-scale variability "uncertainty". I believe this is mis-leading. The newly added paragraph does not improve the assessment much. The authors write in this paragraph "… the standard deviation of all Level 2 observations within a given 1x1 degree box … should in any case be larger than the uncertainty derived from error propagation". This would be the case if all errors are statistical errors. However, it is clear that there may be substantial systematic errors in the retrieval as well.
> The authors write that because the subscale standard deviation is larger than the assessed propagated retrieval error, data users may be more interested in this quantity. I believe, in turn, that users should be able to tell apart what is variability and what is error. Only the latter is an actual uncertainty, the former should be a measure of a meaningful physical behaviour.

> As such, I maintain my previous remark that if the authors choose to report subscale variability, they should call it "subscale variability". If the authors choose to do more rigorous error analysis, more detail is needed on how the error is computed and how it is propagated. In any case, the reader needs more details on how exactly Fig. 8 (top panel) was created.

**Reply**: As mentioned above, we have added detail as to how Fig. 8 (top panel is created) and maybe that clarifies the issue already.

We further believe there is some confusion here about the term 'sub-scale variability' and 'uncertainty'. We cannot agree with the reviewer's premise that we 'choose to report subscale variability' as a measure of uncertainty.

When we talk about 'sub-scale' in the paper, this term refers to inhomogeneity of clouds at scales smaller than individual MODIS observations. We have re-examined the occurrence of the term 'sub-scale' in our paper. The terminology is clarified on Page 2 as well as in Section 3.3 and is consistent also with the literature cited in Section 3.3. It is not clear to us how and if the reviewer separates this scale from the 1x1 degree scale at which the CDNC climatology is reported. That may be the reason for this apparent mis-understanding.

The term 'uncertainty' in our paper refers to the following: In our paper we describe a monthly mean climatology of CDNC at 1x1 degree spatial resolution. When we try to characterize the uncertainty of this product, we need to know the combined effect of retrieval uncertainties and how variable CDNC is temporally and spatially within this 1x1 degree box. We argue (correctly, in our view) that the uncertainty of the monthly mean CDNC estimates is driven mostly by synoptic-scale variability (except for some Sc regions). We further argue (correctly, in our view) that the total uncertainty is best captured by the standard deviation. We further argue (correctly, in our view) that this standard deviation encapsulates also random uncertainties in the retrieval process. Reviewer 2 appears to also agree with this.

We do agree with the reviewer that it would be nice to separate true natural variability from uncertainties that are caused purely by retrieval noise. In the context of a different cloud product, have made an attempt at disentangling those two components. Because of the complexity introduced by that method, that approach is beyond the scope of the current paper, but we refer the reviewer to that relevant paper which is currently under review in AMT:

> Stengel, M., Stapelberg, S., Sus, O., Schlundt, C., Poulsen, C., Thomas, G., Christensen, M., Carbajal Henken, C., Preusker, R., Fischer, J., Devasthale, A., Willén, U., Karlsson, K.-G., McGarragh, G. R., Proud, S., Povey, A. C., Grainger, D. G., Meirink, J. F., Feofilov, A., Bennartz, R., Bojanowski, J., and Hollmann, R.: Cloud property datasets retrieved from AVHRR, MODIS, AATSR and MERIS in the framework of the Cloud_cci project, Earth Syst. Sci. Data Discuss., https://doi.org/10.5194/essd-2017-48, in review, 2017.

Another point made by both reviewers relates to systematic errors. This point is addressed at the very top of this reply.

**2.2. Minor comments**

**Reply: ALL FIXED**
- p2 l28: km²
- p16 l18: consistent
- p16 l26: "These … themselves"
- Caption Fig. 3: "fraction of a typical" - the authors should clarify what they mean by "typical"

**3. Reviewer 2**

**3.1. Major comments**

**3.1.1. Uncertainty analysis**

**Reviewer Comment:**

The new section does somewhat clear up the matter of the uncertainty vs variability, particularly for the stratocumulus regions where the overall variability (i.e. actual variability + instrument uncertainty) is smaller than the instrument uncertainty alone. It is now helpful that the meaning of the uncertainty is made clear in the paper.

Although, this raises the issue of systematic errors/offsets vs random uncertainties. The use of the instrument uncertainties from MODIS (which are just the radiance uncertainties propagated through to reff and tau) neglects uncertainties in the forward model relating to heterogenetiy, etc. This might produce a fairly constant offset error in Nd, which would therefore not show up as variability in the standard deviation. In the current paper the propagation of errors in reff, etc. through to Nd that was done in Bennartz (2007) is mentioned – such an analysis might do a better job of estimating such offset errors to give a better estimate of how far off the quoted Nd values are from reality. However, they don't seem to be used in the current work. I think that it would be good to mention the possibility of such "offset errors" and to quote the uncertainty range calculated in Bennartz (2007).

**Reply**: Please see our reply at the very top. We have followed the reviewer's suggestion and added more information about systematic errors. We also gave more concrete references to the Bennartz (2007) study.

**3.1.2. Solar Zenith Angle dependence**

**Reviewer Comment:**

p.7, L31 – "Conversely, at high solar zenith angle also the the effective radius at 3.7 µm might also be biased leading to possible increases in CDNC by 40% to 70% at solar zenith angles higher than about 70 degrees (Grosvenor and Wood, 2014)."

This doesn't quite address what I was trying to convey with my review comment. Firstly, Grosvenor and Wood (2014) showed that optical depth biases were mainly responsible for the 40-70% increase in CDNC. I think that a sentence like this should be moved to p.9, L15 where you talk about the view geometry biases. There, instead of "Grosvenor and Wood (2014) address the dependency of cloud microphysical retrievals on solar zenith angle. ", you could perhaps write :-

 "Grosvenor and Wood (2014) address the dependency of cloud microphysical retrievals on solar zenith angle demonstrating a possible increases in CDNC by 40% to 70% at solar zenith angles higher than about 70 degrees."

Then on p.7 something like this would be more akin to what I meant "However, at high solar zenith angles Grosvenor and Wood (2014) demonstrated that resolved (as opposed to sub-pixel) 3D radiative effects are likely to cause the effective radius to be

biased high, with larger biases expected for the 3.7 μm retrieval compared to the 1.6 μm one."

**Reply**: We have followed these suggestions and added the suggested wording.

**3.1.3. Sampling in regional boxes**

**Reviewer Comment:**

From your response:-

"We are not too concerned with the data density. Recall that only grid-boxes with at least ten days per month with at least 10 observations each make it into the climatology. Those were then averaged to get the average values for each of the larger boxes like X12. Thus each box has virtually hundreds of observations in it."

However, there may be few 1x1 degree grid boxes that are included in the region X12 for a given month, and this will vary by month. It would be useful to check what these numbers look like and to quote them in the paper.

**Reply**: In response to the reviewer's comments we have now revised Figures 5 and 6, included the percentage of valid grid-points within each region, and added discussion in Section 4.2.1 and 4.2.2. We agree this was helpful as it helped clarify the differences in annual cycle better.

**3.1.4. Issues RE "Specific comments"**

**Reviewer Comment:**

p.3, L19 – "as ultimately one would be interested in the number of cloud droplets activated at cloud base and not the number of cloud droplets observed" – I think this would depend on the application. Some studies may be interested in how the cloud top CDNC might change due to lateral mixing, evaporation, etc., or removal of CDNC by precipitation and not necessarily just the cloud base CDNC. I can see that the cloud base CDNC would be of interest for comparing to model processes, but I think that the statement here generalizes too much.

This does not seem to have been addressed.

**Reply**: Correct. Should be fixed now.

**3.2. Minor comments**

**Reply: ALL FIXED**

- p.5, L6 - "MODI derived".
- p.6, L11 – No need for the commas here.
- P13, L7 – "Out lined"
- Fig. 8 seems to have no caption.
- p.16, L18 – "consisntent"

- p.16, L24 – "our artefacts"
- p.16, L26 – "itself" should be "themselves".
- P17, L4 - "by directly" should be "directly by"
- p.16, L21 :- "Further retrieval issue might arise at high zenith angles and/or near the ice edge. At high zenith angles, earlier work by Grosvenor and Wood (2014) showed the effective radius at 3.7 μm to be more strongly biased, potentially leading to retrieval issues."
    - should include "solar" before "zenith" in both instances. p.12, L25 – "observe" should be "observed".

**Global and regional estimates of warm cloud droplet number concentration based on 13 years of AQUA-MODIS observations**

Ralf Bennartz[1,2], John Rausch[2]

[1] Earth and Environmental Science Department, Vanderbilt University, Nashville, TN, 37240, USA
[2] Space Science and Engineering Center, University of Wisconsin – Madison, WI, 53706, USA

*Correspondence to*: Ralf Bennartz (ralf.bennartz@vanderbilt.edu)

**Abstract.** We present and evaluate a climatology of cloud droplet number concentration (CDNC) based on 13 years of Aqua-MODIS observations. The climatology provides monthly mean 1x1 degree CDNC values plus associated uncertainties over the global ice-free oceans. All values are in-cloud values, i.e the reported CDNC value will be valid for the cloudy part of the grid-box. Here, we provide an overview on how the climatology was generated and assess and quantify potential systematic error sources including effects of broken clouds, and remaining artefacts caused by the retrieval process or related to observation geometry. Retrievals and evaluations were performed at the scale of initial MODIS observations (in contrast to some earlier climatologies, which were created based on already gridded data). This allowed us to implement additional screening criteria, so that observations inconsistent with key assumptions made in the CDNC retrieval could be rejected. Application of these additional screening criteria led to significant changes in the annual cycle of CDNC both in terms of its phase and magnitude. After an optimal screening was established a final CDNC climatology was generated. Resulting CDNC uncertainties are reported as monthly-mean standard deviations of CDNC over each 1x1 degree grid box. These uncertainties 
[revised manuscript text omitted]
 recent studies have made progress toward understanding and possibly correcting the impact of this variability on cloud remote sensing (Zhang et al., (2012), Zhang et al. (2016) and references therein). Furthermore, the cloud microphysical interpretation of retrieved CDNC is also not straightforward. Entrainment mixing processes,

20 precipitation formation, and additional activation above cloud base can lead to differences between the number of cloud droplets activated at cloud base and the number of cloud droplets observed. Studies interested in activation of cloud droplets at cloud base would need to take these differences into account. We will elaborate more on issues related to three-dimensional cloud structure as well as cloud microphysical assumptions in Section 3.

The remainder of this paper is structured as follows. In Section 2 we briefly describe the datasets as well as the principal

25 methods used here to derive CDNC from satellite observations. Section 3 summarizes issues related to the use of the ISBLC as well as other assumptions made in the retrieval process, thereby providing interpretational context for the use of satellite-derived CDNC retrievals as well as guidance on the expected magnitude and relative importance of uncertainties introduced by the various assumptions. In Section 4 we address actual uncertainties and possible biases in CDNC retrievals. Potential biases are for example caused by remaining artefacts in the underlying MODIS retrievals caused by unresolved dependencies

30 on observation geometry or underlying assumptions on the width of the droplet spectrum. In addition, in Section 4 we also validate our CDNC retrievals against in-situ observations of CDNC taken during the VOCALS-Rex campaign and summarized by Painemal and Zuidema (2011). In Section 5 we evaluate the 13-year climatology of MODIS observations. In our analysis we put particular emphasis on the phase and amplitude of the observed annual cycle of CDNC over various

regions of the globe. We further identify areas where trends in CDNC are observed. In Section 6 we provide concluding remarks and discussion of remaining issues that could help improve future satellite-based CDNC estimates.

**2. Datasets and Methods**

**2.1. MODIS Collection 6 cloud parameters**

5 Observational data are from NASA's MODIS Collection 6 (C6) Level-2 Cloud Product (Platnick et al., 2015;Platnick et al., 2017). The term 'Level-2' refers to individual MODIS cloud retrievals at a resolution of 1x1 km at nadir. The Level-2 Cloud Product provides retrievals of cloud optical thickness, cloud top temperature, and three droplet effective radii retrievals using radiances observed at 1.6, 2.1 and 3.7 μm. These cloud parameter retrievals form the basis of our CDNC retrievals. Relative to the earlier MODIS Collection 5, there are several improvements in retrievals of parameters necessary to determine CDNC.

10 These improvements include a better co-registration of the visible and near-infrared focal planes of the Aqua-MODIS instrument as well as significant improvements in the forward radiative transfer models used in the retrieval framework (Platnick et al., 2015). The impact of these changes on CDNC retrievals and gridded climatologies is assessed in detail in Rausch et al. (2016). For the current study, Level-2 cloud retrievals are from Aqua MODIS spanning the years 2003 through 2015. Additionally, for some comparisons with in-situ observations, selected Terra-MODIS granules were used also.

15 ### 2.2. Derivation of CDNC

Under the ISBLC assumption, closed formulas can be derived that relate between two different pairs of cloud physical variables. The first pair of variables are cloud optical depth and effective radius at cloud top, and the second pair of variables are CDNC and cloud geometrical thickness (Brenguier et al., 2000). Following the notation of Bennartz (2007), these relations are:

$$W = \frac{5}{9}\rho_l \tau r_{e,top} = \frac{1}{2}c_w H^2 \qquad (1)$$

where $W$ is the liquid water path, $\rho_l$ the density of liquid water, $\tau$ the optical depth, $r_{e,top}$ the effective radius at the top of the idealized ISBLC, $c_w$ the condensation rate, and $H$ the ISBLC's geometrical thickness. The second relation provides an

25 estimate for CDNC:

$$N = \frac{\tau^3}{k}[2W]^{-5/2}\left[\frac{3}{5}\pi Q\right]^{-3}\left[\frac{3}{4\pi\rho_l}\right]^{-2}c_w^{1/2} \qquad (2)$$

where N represents CDNC, $k$ is a factor related to the dispersion of the assumed cloud droplet size distribution, and $Q$ is the scattering efficiency of the cloud droplets. The variable k exhibits some variability (Brenguier et al., 2011;Martin et al., 1994) but is set constant at a value of $k$=0.8 here. A realistic uncertainty estimate for k between different studies is about 20%. Similarly, $Q$ is set to its geometric optics limit value of $Q$=2. Bennartz (2007) shows that uncertainties in the representation of $Q$ and $k$ are only a minor contributor to the total uncertainty in $N$ and $H$. The condensation rate $c_w$ is calculated as 80% of its maximum adiabatic value for the MODIS-derived cloud top temperature (CTT). In summary, our particular implementation of the ISBLC for the climatology presented here is CDNC Constant vertically, $k$=0.8, $Q$=2, and $c_w$=0.8*f(CTT), where $f(CTT)$ is the full adiabatic condensation rate as function of cloud top temperature (at a pressure assumed fixed at 850 hPa).

[revised manuscript text omitted]

and Wood (2014) demonstrated that resolved (as opposed to sub-pixel) 3D radiative effects are likely to cause the effective radius to be biased high, with larger biases expected for the 3.7 µm retrieval compared to the 1.6 µm one.

The blue line depicted in Figure 1 highlights the issue. Even without accounting for the three-dimensional radiative transfer, partly cloudy scenes will exhibit positively biased effective radii. The blue cross in Figure 1 corresponds to an ISBLC with N=500 cm$^{-3}$ and H=175 m. The blue line gives the resulting observed reflectance, if this cloud covers between 0 and 100 % of the sensor's field of view. For all three wavelengths the blue line cuts through the retrieval grid toward larger effective radii and smaller CDNC. Figure 3 (right panels) shows the impact on retrieved effective radii as one moves from completely cloudy to nearly cloud-free. For sub-pixel fractions of open water above 10% the retrieved effective radius at 1.6 µm starts to exceed 2.1 and 3.7 µm but all three effective radii increase significantly as the pixel becomes less cloud-filled. Zhang et al. (2012b) and Hayes et al. (2010) discuss this effect in more detail, which is also present in case of sub-scale inhomogeneities of fully cloud covered observations (i.e. if cloud thickness and liquid water path varies within the field-of-view). Shading and side-illumination of broken clouds as well as true three-dimensional radiative transfer effects modify this picture somewhat but at the same time the inherent averaging performed over a 1 x 1 km$^2$ MODIS field-of-view averages out some of these higher-order effects (Zhang et al., 2012b). Because of the strong impact of sub-scale inhomogeneity we have limited the climatology to cases where $r_{e,3.7} > r_{e,2.1} > r_{e,1.6}$ , so that in the example shown in Figure 3 (right panels) the largest part of the retrievals would be (correctly) rejected. We discuss the impact of this criterion on the actual climatology in Section 4.

**3.4. Aerosol above clouds**

[revised manuscript text omitted]

**4.2.1. Impact of stratification**

5  Figure 5 shows the mean relative difference 'Stratified' versus 'Non-Stratified' for the year 2008. In the top panel only 1x1 degree grid boxes are shown where all 12 months have valid values for both, 'Stratified' and 'Non-Stratified'. In particular in areas where broken clouds are frequently observed, the stratification criterion removes a significant number of observations, sometimes in the order of 90 %. Resulting differences in climatology are in general positive with the exception of a narrow band in the tropical Pacific. Typically relative differences in mean CDNC climatology are smaller than 10 %

10  with a few exceptions off the east coasts of Asia and North America.

Figure 5 also shows the (2008) CDNC annual cycle for four selected areas. We selected areas where the difference between Stratified' and 'Non-Stratified' are particularly large. The plots labelled PVG ('Percentage of Valid Grid-boxes') in Figure 5 further show the percentage of valid 1x1 degree grid-boxes in each region depending on which stratification criterion was used. The three different estimates of CDNC given in each of the regional mean CDNC panels in Figure 5 can be interpreted

15  as follows: Deviations between the blue and the red curves reflect differences in CDNC estimates for grid-boxes where both 'stratified' and 'non-stratified' provide valid CDNC values. In contrast, deviations between the red and the black curves include the additional effect of 'non-stratified' results being available in more grid-boxes, thereby depending on PVG differences (red versus black), the black curve samples more grid-boxes within a given region. For example, only about 10 % of the grid-boxes in the region X12 provide valid CDNC values in the 'stratified' case, whereas for 'non-stratified' about 90

20  % of the grid-boxes provide valid CDNC values. In this same region (X12) one can identify a secondary peak in 'non-stratified' the annual cycle in October that does not exist, if only grid-boxes that hold valid stratified observations enter the regional averages. Thus, the apparent difference between the red and the blue curve is caused by differences between the grid-box-averaged retrieved values for 'stratified' and 'non-stratified' and not just by 'non-stratified' seeing a different and larger sub-region of X12. The area BC1 shows an annual cycle offset by two months between the two climatologies. Similar

25  to the secondary peak seen in X12, this offset in annual cycle is not caused by different grid-boxes within the region being populated, as the black and the blue curve are nearly on top of each other.

In contrast, the area R05 shows in general much lower CDNC and also a decreased annual cycle, if the stratification criterion is not applied (black versus red curve). However, if the dataset is limited to only those areas, where stratified results are available, these differences are less pronounced (blue versus red curve), indicating that the differences seen between the

30  black and red curves for R05 are largely caused by 'non-stratified' sampling a larger sub-area of R05. In summary, we find that deviations in mean value between 'Stratified' and 'Non-Stratified' are small but largely systematic globally. Further, the annual cycle of CDNC can be strongly affected, depending on the area observed.

**4.2.2. Impact of retrieval artefacts**

Figure 6 shows the impact of the above discussed retrieval artefacts on the CDNC climatology. The interpretation of the different curves in Figure 6 is similar to the interpretation discussed for Figure 5 in Section 4.2.1, except that Figure 6 compares the 'stratified' results used in the final climatology with 'flagged results' in which sampling was only performed

5   for observation geometries that did not show large variations in retrieved effective radius and optical depth (see Section 4.1 and Table 1). The additional constraints on observation geometry significantly reduce spatial coverage as can be seen in the upper panel of Figure 6 as well as in the PVG plots for the individual regions also shown in Figure 6. The relative difference in CDNC between 'flagged' and 'stratified' is less than 10 %. The annual cycles for the four selected areas also discussed under 4.2.1. are very similar between 'flagged' and 'stratified'. An exception to this is R03 where the 'flagged' results show

10  an enhanced annual cycle over the 'stratified' results but does not show a shift in annual cycle. This enhanced annual cycle appears to be caused solely by the difference in coverage between 'flagged' and 'stratified'. If the averages for R03 are confined to grid-boxes where flagged results are also available (blue curve), those are nearly identical again to the red curve. Other selected areas (not shown) substantiate these findings, i.e. the relative differences between 'flagged' and 'stratified' are small, the annual cycle appears to be not affected, but the magnitude of the annual cycle is somewhat muted in the

15  'stratified' cases (caused by differences in the number of valid 1x1 deg grid boxes per region between 'flagged' and 'stratified'). Clearly, while some of the retrieval artefacts discussed in Section 4.1 propagate through into the climatology, their impact is somewhat reduced by averaging out the strong observation geometry dependencies seen in the Level 2 data (Section 4.1.) when the monthly 1x1 degree mean values are aggregated. Recall also that the retrieval artefacts cannot easily be corrected for without potentially re-deriving new Level-2 cloud optical properties retrievals. While a restriction of the

[revised manuscript text omitted]

**4.4 Uncertainty estimates**

As outlined in Section 2.3, two main pathways exist for the definition of uncertainties associated with the climatology. Firstly, uncertainties in retrieved effective radius and optical depth reported in the MODIS Collection 6 Level 2 product can be propagated forward to yield Level 2 uncertainties in CDNC. All auxiliary parameters in the retrieval (i.e. $c_w$, Q, and k in Eq. (2)) were assigned the same uncertainties as in Bennartz (2007). That is, $k=0.8\pm0.1$, $Q=2\pm0.1$, $c_w=(0.8\pm0.1)*f(CTT)$. We note here that these parameters as well as the MODIS Level-2 uncertainties reported for effective radius and optical depth retrievals are not fully random and might contain bias components that are difficult to quantify without independent validation information. The uncertainties calculated via error propagation therefore do not strictly separate the effect of

random versus systematic errors. We note further that the combined contribution of $k$, $Q$, and $c_w$ to the total uncertainty in CDNC is only around 15 % (Bennartz, 2007), so that retrieval errors in optical depth and effective radius dominate the error budget for CDNC. Furthermore, some possible error sources are not included in the MODIS Level-2 effective radius and optical depth reported in the MODIS Level 2 Collection 6 data (e.g. the effects of sub-scale inhomogeneity are not included

5   (Platnick et al., 2017)). Therefore, while instructive, CDNC uncertainties based on error propagation might underestimate the true uncertainty even of individual Level-2 retrievals.

A second way of characterizing uncertainty in the monthly mean CDNC estimates is to calculate the standard deviation of all Level 2 observations within a given 1x1 degree box. The standard deviation would include the synoptic variability of CDNC as well as all actual random retrieval uncertainties and should in any case be larger than the uncertainty derived from error

10  propagation. However, the standard deviation would fail to capture any systematic errors that are consistent throughout the spatial averaging range (1x1 degree) and time period (1 month). Uncertainty estimates based on standard deviation therefore also likely underestimate the true variability of the mean CDNC. However, in contrast to the uncertainty estimates based on error propagation, they fully account for all truly random sources of variability including e.g. random fluctuations in cloud sub-scale inhomogeneity at the individual pixel level.

15  Here, we compare these two measures (error propagation versus standard deviation). Recall from Section 2.3 that the monthly uncertainties for each grid-box were calculated as the square-root of the mean of the daily uncertainties over the course of each month, thereby following the approach used in NASA's Level-3 gridded MODIS cloud product (Hubanks et al., 2016). Figure 8 compares the two different uncertainty estimates in terms of the relative uncertainty averaged over the entire year 2008. Except for some stratocumulus regions the uncertainty estimate based on the standard deviation of the

20  individual Level 2 observations is significantly larger than the uncertainty based on error propagation. This would be consistent with monthly uncertainties driven by synoptic variability.  However, for some of the stratocumulus areas, outlined with black isolines in the bottom panel of Figure 8, the uncertainty by standard deviation is smaller  (by up to 50%) than the uncertainty by forward propagation of retrieval errors.  Hypothetically, and ignoring systematic errors in the discussion of error propagating, even if the actual CDNC in those areas was perfectly constant spatially and temporarily, the uncertainty

25  by standard deviation would be identical to the uncertainty by forward-propagated retrieval error. The only scenario under which this situation could be reversed would be, if estimated uncertainties feeding into the theoretical forward propagation are too large. For example, if the reported uncertainty of the effective radius in the MODIS Level 2 product were too large, this would lead to too large uncertainties in the propagated CDNC uncertainty. Thus, from the bottom panel of Figure 8, it would appear that the forward propagated uncertainties for individual CDNC retrievals are slightly too large. That is, we put

30  slightly less trust into the Level 2 observations than appears to be warranted by this comparison.

For the practical matter of characterizing uncertainty associated with CDNC retrievals for the gridded product, the reported uncertainty estimates should include the impact of day-to-day variability in order to allow for a realistic estimate of the true variability of CDNC.  In the final CDNC climatology we therefore report the standard deviation of the retrieved CDNC values within each 1x1 degree grid box as an uncertainty estimate. It is important to note that this measure of uncertainty

only addresses random errors and does not address possible systematic errors. Such possible systematic errors are better discussed in the framework of sensitivity studies as for example outlined in Section 4.2.2 for the case of retrieval artefacts.

**5. Climatological results**

**5.1. Global overview**

[revised manuscript text omitted]

CDNC reported for the area BB1 and the period 1982 - 2010 (Bennartz et al., 2011). In that earlier publication we have also established a causal link between the Chinese $SO_2$ emissions and wintertime CDNC over the East China Sea. We speculate that decreases in Chinese $SO_2$ emissions over the last decade or so have partly reversed the effect seen in the original study Bennartz et al. (2011). This speculation is substantiated by other studies that report a slight decrease in Chinese $SO_2$

5   emissions from its peak value in the 2004 to 2006 time range, the decrease in $SO_2$ emissions largely caused by flue gas desulfurization technology installed in coal power plants (Lu et al., 2011). Satellite observations suggest an even larger decrease in Chinese $SO_2$ emissions by 50 % for the recent time period 2012-2015 compared to the year 2005 (Krotkov et al., 2016).

**6. Conclusions**

10  The climatology described in this publication provides a number of incremental improvements over earlier climatologies published by us. Importantly, by making the climatology static and providing a DOI, we hope to be able to contribute to a better traceability of results through different studies that might use this climatology. We believe this is a particularly significant issue, as design choices in the generation of the climatology, such as data screening, can have a major influence on scientific results. To this extent we have aimed at clarifying as accurately as possible the screening choices as well as any

15  other decisions that went into the generation of this climatology. For example, we decided to screen out any observation where the three MODIS-derived effective radii are inconsistent with the underlying assumption of the ISBLC. While this additional screening reduces data coverage, it at least partly guards against the misinterpretation as variability in CDNC of three-dimensional radiative transfer effects in broken clouds. We further show that neglecting this screening does not only lead to moderate differences in the annually averaged CDNC but also to sometimes large shifts in phase and amplitude of the

20  annual cycle of CDNC in various regions.  Potentially adverse effects of this screening could occur in situations where the three effective radii are very close to each other, as might be the case for thin stratocumulus clouds. In such cases random noise in the observations might remove valid observations and potentially bias CDNC slightly low as preferably higher effective radii at 3.7 μm would be selected. This hypothesis would be consistent with our finding comparing to the PZ11 cases. However, we believe this issue secondary compared to the large number of situations where the radiation field is

25  affected by either broken clouds or precipitation and the three effective radii clearly indicate that key assumptions in the CDNC retrieval are violated. Further retrieval issue might arise at high solar zenith angles and/or near the ice edge. At high solar zenith angles, earlier work by Grosvenor and Wood (2014) showed the effective radius at 3.7 μm to be more strongly biased, potentially leading to retrieval issues. Further, cloud mask classification errors near the ice edge might lead to increased noise and artefacts in those regions.

30  We found some remaining retrieval artefacts in the MODIS-retrieved effective radius and optical depth that propagate through into artefacts of the CDNC climatology as well. These artefacts manifest themselves in a strong dependency of retrieved effective radius on scattering angle and could potentially be related to the treatment of first order scattering in the

MODIS Level 2 retrievals of effective radius and optical depth. Our assessment of those effects on the climatology shows however that effects clearly visible in the Level-2 data are averaged out to some degree in the aggregated climatology. We therefore did not in the final climatology include any additional screening for those effects. However, we feel that it might be beneficial in future work to re-create retrievals working directly on the Level-1 reflectances that address some of the issues

5   we have identified. Such retrievals could be specific to stratiform boundary layer clouds and might include more appropriate assumptions for example about the width of the droplet spectrum and the stratification of the cloud as well as a finer retrieval grid or a direct calculation of the first one or two orders of scattering that corresponds to the actual observation geometry for each pixel. However, other error sources, for example related to observation geometry or broken clouds, cannot be resolved directly by retrievals of CDNC from Level 1. Another important issue not addressed here is the consistency between Terra-

[revised manuscript text omitted]